# Pre-trained multimodal large language model enhances dermatological diagnosis using SkinGPT-4

Juexiao Zhou[1,2,3,8], Xiaonan He[4,8] ✉, Liyuan Sun[5,8], Jiannan Xu[5], Xiuying Chen[1,2], Yuetan Chu [1,2], Longxi Zhou [1,2], Xingyu Liao [1,2], Bin Zhang[1,2], Shawn Afvari[3,6,7] & Xin Gao [1,2] ✉

Large language models (LLMs) are seen to have tremendous potential in advancing medical diagnosis recently, particularly in dermatological diagnosis, which is a very important task as skin and subcutaneous diseases rank high among the leading contributors to the global burden of nonfatal diseases. Here we present SkinGPT-4, which is an interactive dermatology diagnostic system based on multimodal large language models. We have aligned a pre-trained vision transformer with an LLM named Llama-2-13b-chat by collecting an extensive collection of skin disease images (comprising 52,929 publicly available and proprietary images) along with clinical concepts and doctors' notes, and designing a two-step training strategy. We have quantitatively evaluated SkinGPT-4 on 150 real-life cases with board-certified dermatologists. With SkinGPT-4, users could upload their own skin photos for diagnosis, and the system could autonomously evaluate the images, identify the characteristics and categories of the skin conditions, perform in-depth analysis, and provide interactive treatment recommendations.

Skin and subcutaneous diseases rank as the fourth major cause of nonfatal disease burden worldwide, affecting a considerable proportion of individuals, with a prevalence ranging from 30 to 70% across all ages and regions[1]. However, dermatologists are consistently in short supply, particularly in rural areas, and consultation costs are on the rise[2–4]. As a result, the responsibility of diagnosis often falls on non-specialists such as primary care physicians, nurse practitioners, and physician assistants, which may have limited knowledge and training[5] and low accuracy on diagnosis[6,7]. The use of store-and-forward tele-dermatology has become dramatically popular in order to expand the range of services available to medical professionals[8], which involves transmitting digital images of the affected skin area (usually taken using a digital camera or smartphone)[9] and other relevant medical information from users to dermatologists. Then, the dermatologist reviews the case remotely and advises on diagnosis, workup, treatment, and follow-up recommendations[10,11]. Nonetheless, the field of dermatology diagnosis faces three significant hurdles[12]. First, there is a shortage of dermatologists accessible to diagnose patients, particularly in rural regions. Second, accurately interpreting skin disease images poses a considerable challenge. Lastly, generating patient-friendly diagnostic reports is usually a time-consuming and labor-intensive task for dermatologists[4,13].

[1]Computer Science Program, Computer, Electrical and Mathematical Sciences and Engineering Division, King Abdullah University of Science and Technology (KAUST), Thuwal, Kingdom of Saudi Arabia. [2]Computational Bioscience Research Center, Computer, Electrical and Mathematical Sciences and Engineering Division, King Abdullah University of Science and Technology (KAUST), Thuwal, Kingdom of Saudi Arabia. [3]DermAssure, LLC, New York, NY, USA. [4]Emergency Critical Care Center, Beijing AnZhen Hospital, Affiliated to Capital Medical University, Beijing, China. [5]Department of Dermatology, Beijing AnZhen Hospital, Affiliated to Capital Medical University, Beijing, China. [6]Department of Dermatology, Brigham and Women's Hospital, Harvard University, Boston, MA, USA. [7]School of Medicine, New York Medical College, Valhalla, NY, USA. [8]These authors contributed equally: Juexiao Zhou, Xiaonan He, Liyuan Sun. ✉ e-mail: hxndoctor@126.com; xin.gao@kaust.edu.sa

Advancements in technology have led to the development of various tools and techniques to aid dermatologists in their diagnosis[13–15]. For example, the development of artificial intelligence (AI) tools to aid in the diagnosis of skin disorders from images has been made possible by recent advancements in deep learning (DL)[16,17], such as skin cancer classification[18–27], dermatopathology[28–30], predicting novel risk factors or epidemiology[31,32], identifying onychomycosis[33], quantifying alopecia areata[34], classify skin lesions from mpox virus infection[35], and so on[4]. Among these, most studies have predominantly concentrated on identifying skin lesions through dermoscopic images[36–38]. However, dermatoscopy is often not readily available outside of dermatology clinics. Some studies have explored the use of clinical photographs of skin cancer[18], onychomycosis[33], and skin lesions on educational websites[39]. Nevertheless, those methods are tailored for particular diagnostic objectives as classification tasks and their approach still requires further analysis by dermatologists to issue reports and make clinical decisions. Those methods are unable to automatically generate detailed reports in natural language and allow interactive dialogues with patients. At present, there are no such diagnostic systems available for users to self-diagnose skin conditions by submitting images that can automatically and interactively analyze and generate easy-to-understand text reports.

Over the past few months, the field of large language models (LLMs) has seen significant advancements[40,41], offering remarkable language comprehension abilities and the potential to perform complex linguistic tasks. One of the most anticipated models is GPT-4[42], which is a large-scale multimodal model that has demonstrated exceptional capabilities, such as generating accurate and detailed image descriptions, providing explanations for atypical visual occurrences, constructing websites based on handwritten textual descriptions, and even acting as family doctors[43]. Despite these remarkable advancements, some features of GPT-4 are still not accessible to the public and are closed-source. Users need to pay and use some features through API. As an accessible alternative, ChatGPT, which is also developed by OpenAI, has demonstrated the potential to assist in disease diagnosis through conversation with patients[44–49]. By leveraging its advanced natural language processing capabilities, ChatGPT could interpret symptoms and medical history provided by patients and make suggestions for potential diagnoses or referrals to appropriate dermatological specialists[50]. However, it is important to note that most LLMs are limited to text interaction alone currently. Nevertheless, the development of multimodal large language models for medical diagnosis is still in its early stages[51,52], particularly considering the prevalence of image-based data in the field of medical diagnosis, among which, dermatological diagnosis is a very important task but lacks relevant research on enhanced diagnosis with multimodal large language models[53,54].

The idea of providing skin images directly for automatic dermatological diagnosis and generating text reports could greatly help solve the three aforementioned challenges in the field of dermatology diagnosis. However, there exists no method to accomplish this at present. But in related areas, ChatCAD[55] is one of the most advanced approaches that designed various networks to analyze X-rays, CT scans, and MRIs images and generate diverse outputs, which are then transformed into text descriptions. These descriptions are combined as inputs to ChatGPT to generate a condensed report and offer interactive explanations and medical recommendations based on the given image. However, their proposed vision-text models were limited to certain tasks. Meanwhile, for ChatCAD, users need to use ChatGPT's API to upload text descriptions, which could raise data privacy issues[41,56,57] as both medical images and text descriptions contain patients' private information[58–61]. To address those issues, MiniGPT-4[62] is an open-source method that allows users to deploy locally to interface images with state-of-the-art LLMs and interact using natural language without the need to fine-tune both pre-trained large models and

only a small alignment layer. MiniGPT-4 aims to combine the power of a large language model with visual information obtained from a pre-trained vision encoder. To achieve this, the model uses Vicuna[63] as its language decoder, which is built on top of LLaMA[64] and is capable of performing complex linguistic tasks. To process visual information, the same visual encoder used in BLIP-2[65] is employed, which consists of a ViT[66] backbone combined with a pre-trained Q-Former. Both the language and vision models are open-source. To bridge the gap between the visual encoder and the language model, MiniGPT-4 utilizes a linear projection layer. However, MiniGPT-4 is trained on the combined dataset of Conceptual Caption[67], SBU[68], and LAION[69], which are irrelevant to medical images, especially dermatological images. Therefore, it is still challenging to directly apply MiniGPT-4 to specific domains such as formal dermatology diagnosis. Meanwhile, due to the limitations of Vicuna, MiniGPT-4 could not support commercial use, which could also be further improved by incorporating other state-of-the-art large language models.

Inspired by current state-of-the-art multimodal large language models, we present SkinGPT-4, which is an interactive dermatology diagnostic system based on multimodal large language models. (Fig. 1). SkinGPT-4 brings innovation on two fronts. First, SkinGPT-4 is a multimodal large language model aligned with the Llama-2-13b-chat. Second, SkinGPT-4 is a multimodal large language model designed for dermatologic diagnosis. To implement SkinGPT-4, we have designed a framework that aligned a pre-trained vision transformer with a pre-trained large language model named Llama-2-13b-chat. To train SkinGPT-4, we have collected an extensive collection of skin disease images (comprising 52,929 publicly available and proprietary images) along with clinical concepts and doctors' notes (Table 1). We designed a two-step training process to develop SkinGPT-4 as shown in Fig. 2. In the initial step, SkinGPT-4 aligns visual and textual clinical concepts, enabling it to recognize medical features within skin disease images and express those medical features with natural language. In the subsequent step, SkinGPT-4 learns to accurately diagnose the specific types of skin diseases. This comprehensive training methodology ensures the system's proficiency in analyzing and classifying various skin conditions. With SkinGPT-4, users have the ability to upload their own skin photos for diagnosis. The system autonomously evaluates the images, identifies the characteristics and categories of the skin conditions, performs in-depth analysis, and provides interactive treatment recommendations (Fig. 3). Meanwhile, SkinGPT-4's local deployment capability and commitment to user privacy also render it an appealing choice for patients in search of a dependable and precise diagnosis of their skin ailments. To demonstrate the robustness of SkinGPT-4, we conducted quantitative evaluations on 150 real-life cases, which were independently reviewed by board-certified dermatologists (Fig. 4 and Supplementary information). The results showed that SkinGPT-4 consistently provided accurate diagnoses of skin diseases. Though SkinGPT-4 is not a substitute for doctors, it greatly enhances users' understanding of their medical conditions, facilitates improved communication between patients and doctors, expedites the diagnostic process for dermatologists, facilitates triage, and has the potential to advance human-centered care and healthcare equity, particularly in underserved regions[70]. In summary, SkinGPT-4 represents a significant leap forward in the field of dermatology diagnosis in the era of large language models and a valuable exploration of multimodal large language models in medical diagnosis.

## Results
### The overall design of SkinGPT-4

SkinGPT-4 is an interactive system designed to provide a natural language-based diagnosis of skin disease images as shown in Fig. 1. The process commences when the user uploads a skin image, which undergoes encoding by the Vision Transformer (ViT) and Q-Former models to comprehend its contents. The ViT model partitions the

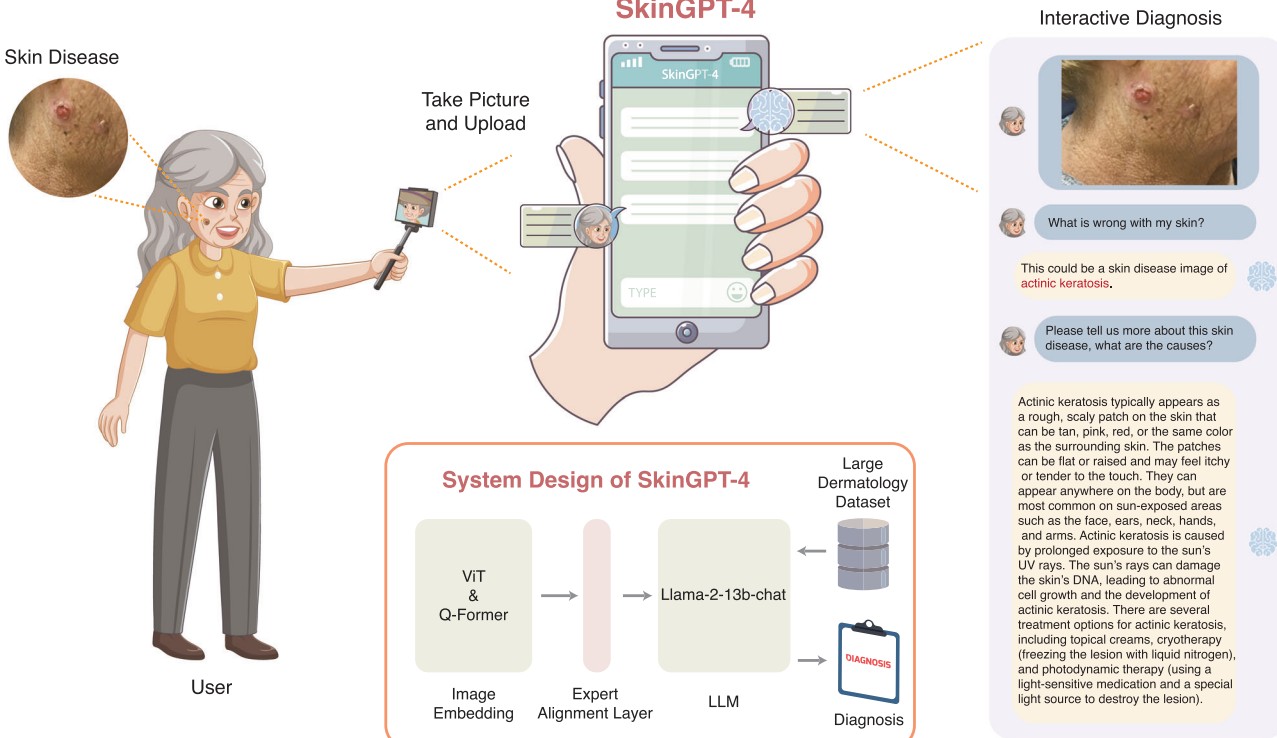

**Fig. 1 | Illustration of SkinGPT-4.** SkinGPT-4 is an interactive dermatology diagnostic system based on multimodal large language models. To implement SkinGPT-4, we have designed a framework that aligned a pre-trained vision transformer with a large language model named Llama-2-13b-chat. SkinGPT-4 was trained on a vast collection (52,929) of both public and in-house skin disease images, accompanied by clinical concepts and doctors' notes. With SkinGPT-4, users could upload their own skin photos for diagnosis, and SkinGPT-4 could autonomously determine the characteristics and categories of skin conditions, perform analysis, provide treatment recommendations, and allow interactive diagnosis. On the right is an example of interactive diagnosis.

image into smaller patches and extracts vital features like edges, textures, and shapes. After that, the Q-Former model generates an embedding of the image based on the features identified by the ViT model, which is done by using a transformer-based architecture that allows the model to consider the context of the image. The alignment layer facilitates the synchronization of visual information and natural language, and the large language model named Llama-2-13b-chat generates the text-based diagnosis. SkinGPT-4 was trained using large skin disease images along with clinical concepts and doctors' notes to allow for interactive dermatological diagnosis. The system could provide an interactive and user-friendly way to help users self-diagnose skin diseases.

### Interactive, informative, and understandable dermatology diagnosis of SkinGPT-4

SkinGPT-4 brings forth a multitude of advantages for both patients and dermatologists. One notable benefit lies in its utilization of comprehensive and trustworthy medical knowledge specifically tailored to skin diseases. This empowers SkinGPT-4 to deliver interactive diagnoses, explanations, and recommendations for skin diseases (Supplementary Movie 1), which presents a challenge for MiniGPT-4. Unlike MiniGPT-4, which lacks training with pertinent medical knowledge and domain-specific adaptation, SkinGPT-4 overcomes this limitation, enhancing its proficiency in the dermatological domain. To demonstrate the advantage of SkinGPT-4 over MiniGPT-4, we presented two real-life examples of interactive diagnosis as shown in Fig. 3. In Fig. 3a, an image is presented of elderly with actinic keratosis on her face. In Supplementary Fig. S1, an image is provided of a patient with eczema fingertips.

For the actinic keratosis case (Fig. 3a), MiniGPT-4 identified features like small and red bumps, and incorrectly diagnosed the skin

disease as acne, while SkinGPT-4 identified features like plaque, nodules, pustules, and scarring, and diagnosed the skin disease as actinic keratosis, which is a common skin condition caused by prolonged exposure to the sun's ultraviolet (UV) rays[71]. During the interactive dialogue, SkinGPT-4 also suggested the cause of the skin disease to be sun exposure, which was also verified by the board-certified dermatologist. For the example case of fingertip eczema (Supplementary Fig. S1), MiniGPT-4 identified some features like cracks and skin flakes but could not accurately diagnose the condition and attributed the cause of the skin disease to dry weather and excessive hand washing. In comparison, SkinGPT-4 identified the features of the skin disease as dry, itchy and flaky skin and diagnosed the type of the skin disease to be fingertip eczema, which was also verified by board-certified dermatologists.

In summary, the absence of dermatological knowledge and domain-specific adaptation poses a significant challenge for MiniGPT-4 in achieving accurate dermatological diagnoses. Contrastingly, SkinGPT-4 successfully and accurately identified the characteristics of the skin diseases displayed in the images. It not only suggested potential disease types but also provided recommendations for potential treatments. This further highlights that domain-specific adaption is crucial for SkinGPT-4 to work for the dermatological diagnosis.

### SkinGPT-4 masters medical features to improve diagnosis with the two-step training

To further illustrate the capability of SkinGPT-4 in enhancing dermatological diagnosis through learning medical features in skin disease images, we conducted ablation studies, as depicted in Fig. 3 by training SkinGPT-4 using either solely the step 1 dataset or the step 2 dataset. As specified in Method and illustrated in Fig. 2, we designed a two-step

**Table. 1 | Characteristics of step 1 dataset**

| Clinical concepts | Number of samples |
|---|---|
| Erythema | 2139 |
| Plaque | 1966 |
| Papule | 1169 |
| Brown (hyperpigmentation) | 759 |
| Scale | 686 |
| Crust | 497 |
| White (hypopigmentation) | 257 |
| Yellow | 245 |
| Erosion | 200 |
| Nodule | 189 |
| Ulcer | 154 |
| Friable | 153 |
| Patch | 149 |
| Dome-shaped | 146 |
| Exudate | 144 |
| Scar | 123 |
| Pustule | 103 |
| Telangiectasia | 100 |
| Black | 90 |
| Purple | 85 |
| Atrophy | 69 |
| Bulla | 64 |
| Umbilicated | 49 |
| Vesicle | 46 |
| Warty/papillomatous | 46 |
| Excoriation | 46 |
| Exophytic/fungating | 42 |
| Xerosis | 35 |
| Induration | 33 |
| Fissure | 32 |
| Sclerosis | 27 |
| Pedunculated | 26 |
| Lichenification | 25 |
| Comedo | 24 |
| Wheal | 21 |
| Flat-topped | 18 |
| Translucent | 16 |
| Macule | 13 |
| Salmon | 10 |
| Purpura/petechiae | 10 |
| Acuminate | 8 |
| Cyst | 6 |
| Blue | 5 |
| Abscess | 5 |
| Poikiloderma | 5 |
| Burrow | 5 |
| Gray | 5 |
| Pigmented | 5 |

It is possible for a single image to have multiple medical concepts at the same time.
The total number of samples is 3886.

training process for SkinGPT-4. Initially, we utilized the step 1 dataset to familiarize SkinGPT-4 with the medical features present in dermatological images and allow SkinGPT-4 to express medical features in skin disease images with natural language. Subsequently, we employed the step 2 dataset to train SkinGPT-4 to achieve a more precise diagnosis of disease types.

In the instance of actinic keratosis (Fig. 3a), SkinGPT-4 trained solely on the step 1 dataset demonstrated its proficiency in identifying pertinent medical features such as plaque, crust, erythema, and umbilication. These precise and comprehensive morphological descriptions accurately captured the characteristics of the skin disease depicted in the image. However, when SkinGPT-4 was exclusively trained on the step 1 dataset, it erroneously diagnosed the skin condition as a viral infection, indicating the importance of incorporating the step 2 dataset for more accurate disease identification. In contrast, when trained solely on the step 2 dataset, SkinGPT-4 failed to capture the accurate morphological descriptions of the skin diseases and instead incorrectly diagnosed it as the result of excessive sebum production. It highlights the necessity of incorporating the step 1 dataset to effectively recognize and comprehend the specific medical features essential for precise dermatological diagnoses. In comparison, SkinGPT-4 with our two-step training simultaneously identified the medical features and diagnosed the skin disease as actinic keratosis. For simple cases such as the fingertip eczema case shown in Supplementary Fig. S1, SkinGPT-4 could also provide more detailed descriptions of the skin disease image, encompass the medical features and accurately identify the type of skin disease. In conclusion, the two-step training process we have implemented allows SkinGPT-4 to effectively comprehend and master medical features in dermatological images, thereby significantly enhancing the accuracy of diagnoses, which is particularly crucial for challenging cases where precise identification of medical features is paramount to accurately determining the type of disease.

### Clinical evaluation of SkinGPT-4 by board-certified dermatologists

To evaluate the reliability and robustness of SkinGPT-4, we conducted a comprehensive study involving a large number of real-life cases (150) and compared its diagnoses with those of board-certified dermatologists. The results, presented in Table 2 and Supplementary information, demonstrated that SkinGPT-4 consistently provided accurate diagnoses that were in agreement with those of the board-certified dermatologists as shown in Fig. 4, as well as in all cases detailed in the Supplementary information.

As shown in Fig. 4a, among the 150 cases, a significant percentage of SkinGPT-4's diagnoses (80.63%) were evaluated as correct or relevant by board-certified dermatologists. This evaluation encompassed both strongly agree (75.00%) and agree (5.63%). In addition, SkinGPT-4's responses regarding the causes of the disease and potential treatments were considered informative (82.50%) and useful (85.63%) by the doctors. Furthermore, SkinGPT-4 proved to be a valuable tool for doctors in the diagnosis process (87.50%) and for patients in gaining a better understanding of their diseases (83.70%). The capability of SkinGPT-4 to support local deployment, ensuring user privacy, garnered high agreement (92.50%), further enhancing the willingness to utilize SkinGPT-4 (77.50%).

Overall, the study demonstrated that SkinGPT-4 delivers reliable diagnoses, aids doctors in the diagnostic process, facilitates patient understanding, and prioritizes user privacy, making it a valuable asset in the field of dermatology.

### SkinGPT-4 acts as a 24/7 on-call family doctor

In comparison to online consultations with dermatologists, which often entail waiting minutes for a response, or in-person consultations with dermatologists, which often entail waiting weeks for an appointment, SkinGPT-4 offers several advantages. First, it is available 24/7, ensuring constant access to medical advice. In addition, SkinGPT-4 provides rapid response times, typically within seconds, as depicted in Fig. 4b, which makes it a swift and convenient option for patients requiring immediate diagnoses outside of regular office hours.

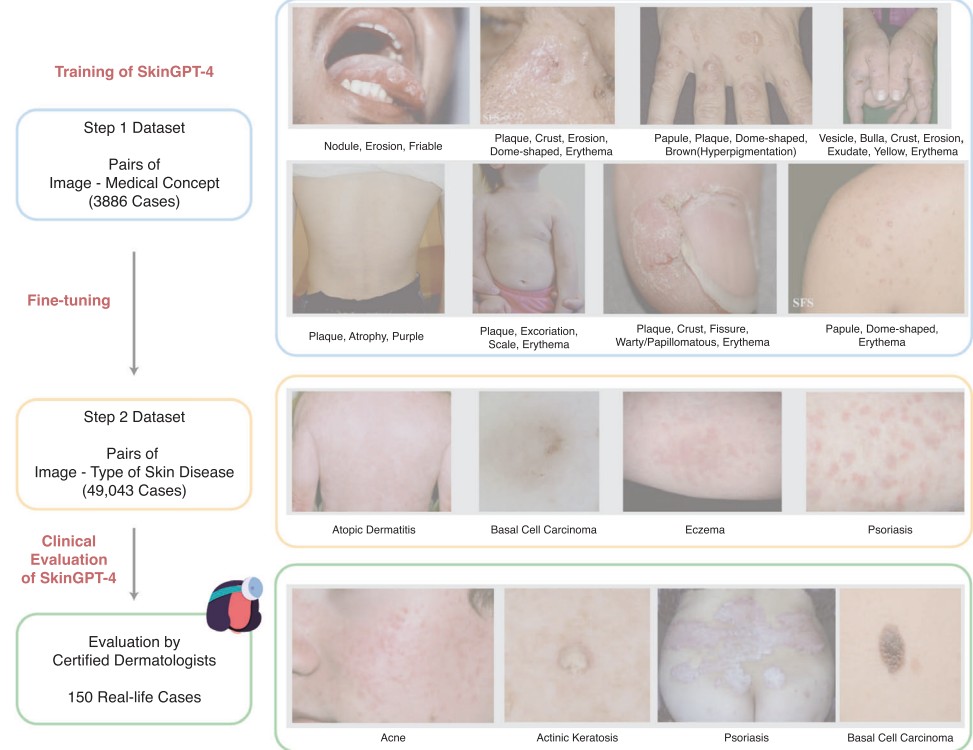

**Fig. 2 | Illustration of our datasets for two-step training of SkinGPT-4.** The notes below each image indicate clinical concepts and types of skin diseases. In addition, we have detailed descriptions from the board-certified dermatologists for images in the step 2 dataset. To avoid causing discomfort, we used a translucent grey box to obscure the displayed skin disease images.

Moreover, SkinGPT-4's ability to offer preliminary diagnoses empowers patients to make informed decisions about seeking in-person medical attention. This feature can help reduce unnecessary visits to the doctor's office, saving patients both time and money. The potential to improve healthcare access is particularly significant in rural areas or regions experiencing a scarcity of dermatologists. In such areas, patients often face lengthy waiting times or must travel considerable distances to see a dermatologist[72]. By leveraging SkinGPT-4, patients can swiftly and conveniently receive preliminary diagnoses, potentially diminishing the need for in-person visits and alleviating the strain on healthcare systems in these underserved regions.

### Consistency of SkinGPT-4's diagnosis

GPT tends to generate results in various formats according to probability and thus the risks and consistency associated with AI-generated content must be carefully considered[73], especially in medical diagnosis. To demonstrate the consistency of the results from SkinGPT-4, we randomly selected 45 samples (5 from each class as depicted in Table 2). For each sample, we conducted 10 independent diagnoses. As shown in Fig. 4c, the diagnoses made on the same graph were consistent with a consistency ratio of 93.73%. For inconsistent cases, features of multiple possible skin types could be observed by board-certified dermatologists, such as the benign tumour could be easily confused with melanoma skin cancer. Overall, the diagnoses of SkinGPT-4 are consistent and reliable.

### Discussion

Our study showcases the promising potential of utilizing visual inputs in LLMs to enhance dermatological diagnosis. With the upcoming release of more advanced LLMs like GPT-4, the accuracy and quality of diagnoses could be further improved. However, it is essential to

address potential privacy concerns associated with using LLMs like ChatGPT and GPT-4 as an API, as it requires users to upload their private data. In contrast, SkinGPT-4 offers a solution to this privacy issue. By allowing users to deploy the model locally, the concerns regarding data privacy are effectively resolved. Users have the autonomy to use SkinGPT-4 within the confines of their own system, ensuring the security and confidentiality of their personal information.

Deploying SkinGPT-4 in real-world scenarios may pose potential challenges, particularly due to the variability in patient-submitted images. Factors contributing to this variability include differences in smartphone camera quality, variations in image pre- and post-processing, diverse angles, and varying lighting conditions. In addition, addressing the diverse severity levels of skin diseases presents another challenge. During the training of SkinGPT-4, we lacked the specific data required to enable the model to identify the severity of skin diseases accurately. Nevertheless, as demonstrated in Supplementary Fig. S2, SkinGPT-4 still exhibits robust and acceptable performance when presented with skin disease images captured under varying angles, lighting conditions, pixel densities, and resolutions with differing severity levels of acne, which were classified according to the Chinese guidelines for the treatment of acne (revised 2019)[74]. As shown in Supplementary Fig. S3, a guideline for users was also implemented, prompting them to capture images as appropriately as possible. This approach aims to standardize the format of uploaded images, facilitating SkinGPT-4's ability to identify skin disease features effectively.

The diagnosis of intricate skin diseases poses an additional challenge for SkinGPT-4. In practice, complex skin diseases frequently occur, encompassing a combination of diverse skin diseases exhibiting a multitude of characteristics. Currently, there is a lack of datasets containing multi-label skin disease images along with corresponding dermatologists' diagnoses. Addressing this gap in data constitutes a

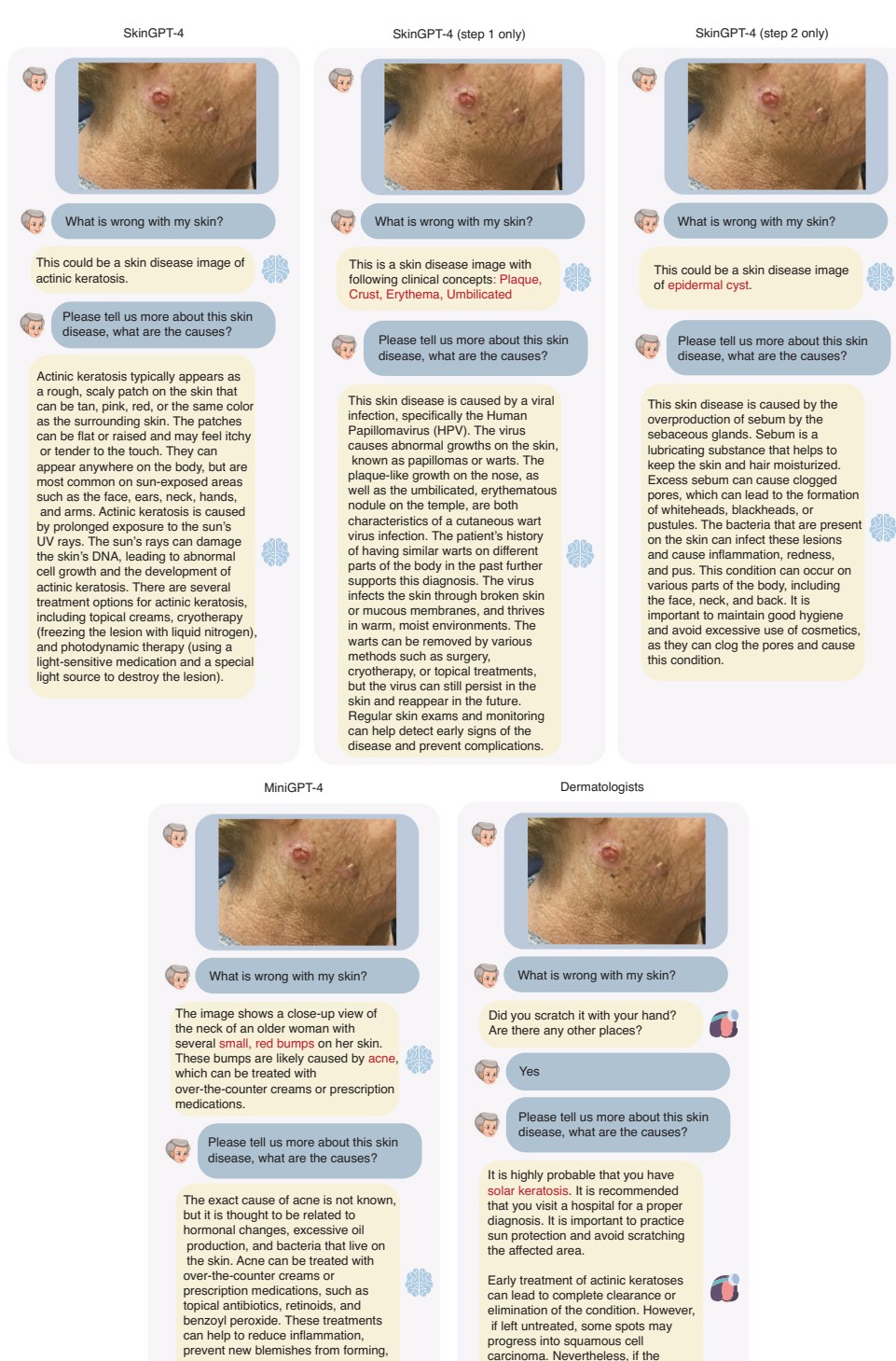

**Fig. 3 | Diagnosis generated by SkinGPT-4, SkinGPT-4 (step 1 only), SkinGPT-4 (step 2 only), MiniGPT-4, and dermatologists.** This figure shows a case of actinic keratosis.

key focus for future research endeavors to apply SkinGPT-4 in the diagnosis of complex skin diseases.

The hallucination of LLMs presents another potential challenge. In the realm of medical diagnosis, the ramifications of misinformation for patients could be fatal. Given that current LLMs are trained on multiple sources, ensuring the absolute accuracy of generated medical facts is an imperative area for further investigation. Potential solutions may entail training more specialized LLMs for medical purposes and implementing iterative diagnostic generation with vote-like mechanisms. This further underscores the role of LLM-based approaches in

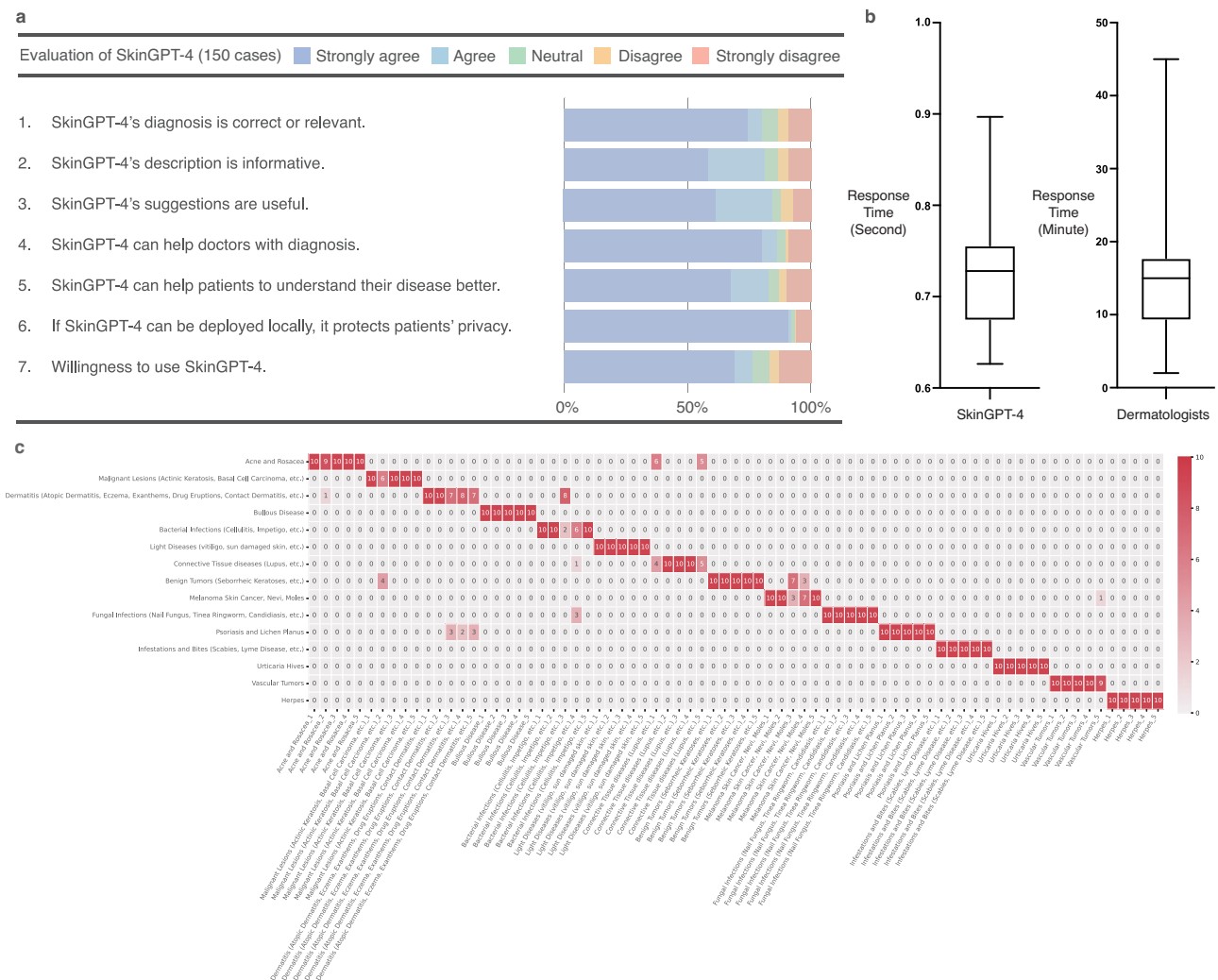

**Fig. 4 | Clinical evaluation of SkinGPT-4 by board-certified offline and online dermatologists. a** Questionnaire-based assessment of SkinGPT-4 by offline dermatologists. The barplot represents the percentage of skin disease cases that the dermatologists agree on. **b** Response time of SkinGPT-4 ($n = 20$) is lower compared to consulting dermatologists online ($n = 20$) (two-tailed Student's $t$ test with $P < 0.00001$). All boxplots represent the first quartile, the median, and the third quartile. The upper whisker indicates the maximum value no further than 1.5 times the interquartile range from the third quartile. The lower whisker indicates the minimum value no further than 1.5 times the interquartile range from the first quartile. Source data are provided as a Source Data file. **c** Consistency test of SkinGPT-4's responses. The x axis indicates test samples, and the y axis indicates diagnostic results.

medicine as tools meant to augment doctors' capabilities in delivering human-centered diagnoses, rather than to supplant them.

Current research on Fitzpatrick V–VI (dark skin tones) is relatively limited, and the performance of state-of-the-art dermatological AI algorithms is notably inferior for lesions on dark skin compared to their efficacy for light-colored skin, especially in cases confirmed by biopsies. The primary challenge arises from the less conspicuous early characteristics of certain dark skin diseases, leading to a more challenging diagnosis[53]. Consequently, individuals with darker skin tones often receive diagnoses at later stages, resulting in heightened morbidity, mortality, and associated costs[75,76]. Compounding this issue is the scarcity of Fitzpatrick V–VI data, such as the Diverse Dermatology Images (DDI) dataset[53], which is insufficient for training deep learning models, particularly those based on LLMs such as SkinGPT-4. In this study, our dataset primarily comprises Fitzpatrick I–IV skin tones, inadvertently limiting the model's efficacy in diagnosing skin diseases in individuals with Fitzpatrick V–VI. To address this limitation, future research endeavors will involve the systematic collection of Fitzpatrick V–VI data and the targeted training of SkinGPT-4 to enhance its diagnostic capabilities for Fitzpatrick V–VI patients.

During the course of a patient's consultation with a dermatologist, the doctor often asks additional questions to gather crucial information that aids in arriving at a precise diagnosis. In contrast, SkinGPT-4 relies on the information provided by users to assist in the diagnostic process. In addition, doctors often engage in empathetic interactions with patients, as the emotional connection could contribute to the diagnostic process. Due to these factors, it remains challenging for SkinGPT-4 to fully replace dermatologists at present. However, SkinGPT-4 still holds significant value as a tool for both patients and dermatologists. It can greatly expedite the diagnostic process and enhance the overall service delivery. By leveraging its capabilities, SkinGPT-4 empowers patients to obtain preliminary insights into their skin conditions and aids dermatologists in providing more efficient care. While it may not fully substitute for the expertise and empathetic nature of dermatologists, SkinGPT-4 serves as a valuable complementary resource in the field of dermatological diagnosis.

As LLMs-based applications like SkinGPT-4 continue to evolve and improve with the acquisition of even more reliable medical training data, the potential for significant advancements in online medical services is enormous. SkinGPT-4 could play a critical role in improving

**Table. 2 | Characteristics of step 2 dataset and clinical evaluation dataset**

| Major classes of skin disease | Number of samples in step 2 dataset | Number of samples in clinical evaluation dataset |
|---|---|---|
| Acne and rosacea | 840 | 10 |
| Malignant lesions (actinic keratosis, basal cell carcinoma, etc.) | 8166 | 10 |
| Dermatitis (atopic dermatitis, eczema, exanthems, drug eruptions, contact dermatitis, etc.) | 5262 | 10 |
| Bullous disease | 448 | 10 |
| Bacterial infections (cellulitis, impetigo, etc.) | 228 | 10 |
| Light diseases (vitiligo, sun-damaged skin, etc.) | 568 | 10 |
| Connective tissue diseases (lupus, etc.) | 420 | 10 |
| Benign tumors (seborrheic keratoses, etc.) | 1916 | 10 |
| Melanoma skin cancer, nevi, moles | 23,373 | 10 |
| Fungal infections (nail fungus, tinea ringworm, candidiasis, etc.) | 2340 | 10 |
| Psoriasis and lichen planus | 3460 | 10 |
| Infestations and bites (scabies, Lyme disease, etc.) | 431 | 10 |
| Urticaria hives | 212 | 10 |
| Vascular tumors | 735 | 10 |
| Herpes | 405 | 10 |
| Others | 239 | / |
| Total | 49,043 | 150 |

access to healthcare and enhancing the quality of medical services for patients worldwide. It is crucial to underscore that no AI system is infallible and entirely free from misinformation and misdiagnosis. Therefore, SkinGPT-4 is not designed to replace dermatologists but rather to serve as an evolving and continuously optimized tool, functioning as an assistant in facilitating communication between patients and doctors. Our aspiration for SkinGPT-4 is to provide patients with more information about skin diseases, while also offering doctors valuable assistance in the diagnostic process. Therefore, we included clear disclaimers and guidance on the software page. This includes a prominent advisory, emphasizing the importance of adhering to medical advice, and a strong recommendation to consult with a qualified physician for specific diagnostic results. These precautionary measures are in place to encourage responsible use and ensure that users comprehend the limitations of the software in a medical context. We will continue our research in this field to further develop and refine this technology.

## Methods

### Ethics

This study employs a deep learning methodology developed through the utilization of publicly available anonymized skin disease images, as well as anonymized in-house skin disease images sourced from hospitals. All research activities strictly adhered to established ethical regulations. The ethics vote for this study was held by the Beijing Anzhen Hospital Ethics Committee, Beijing AnZhen Hospital, affiliated with Capital Medical University in Beijing, China, with ethical approval obtained under referencing ID 2024002X. The ethical approval from the Institutional Biosafety and Bioethics Committee (IBEC), King Abdullah University of Science and Technology was obtained under referencing ID 23IBEC100. For publicly available anonymized skin disease images, we meticulously followed the policies and restrictions outlined by the respective datasets. Consequently, patients' informed consent was not obtained externally. For the in-house dataset, informed consent was collected from all participants whose images are featured in this study. We utilized pre-collected data from the hospital, which is a common practice in dermatological diagnosis. To uphold data anonymity, any sensitive privacy information was systematically removed from the dataset. The use of this particular subset of the data was granted a waiver for informed consent due to its anonymized nature. We did not collect any protected health information (PHI) from the patients who participated in our study, ensuring the confidentiality and privacy of their information.

### Dataset

Our datasets include two public datasets and our private in-house dataset, where the first public dataset was used for the step 1 training, and the second public dataset and our in-house dataset were used for the step 2 training.

The first public dataset named SKINCON[77] is the first medical dataset densely annotated by domain experts to provide annotations useful across multiple disease processes. SKINCON is a skin disease dataset densely annotated by dermatologists and it includes 3230 images from the Fitzpatrick 17k[78] skin disease dataset densely annotated with 48 clinical concepts as shown in Table 1, 22 of which have at least 50 images representing the concept, and 656 skin disease images from the Diverse Dermatology Images dataset[53]. The Fitzpatrick 17k disease annotations lack verification through skin biopsy. In contrast, all DDI diseases underwent verification through skin biopsy. The 48 clinical concepts proposed by SKINCON include Vesicle, Papule, Macule, Plaque, Abscess, Pustule, Bulla, Patch, Nodule, Ulcer, Crust, Erosion, Excoriation, Atrophy, Exudate, Purpura/Petechiae, Fissure, Induration, Xerosis, Telangiectasia, Scale, Scar, Friable, Sclerosis, Pedunculated, Exophytic/Fungating, Warty/Papillomatous, Dome-shaped, Flat-topped, Brown(Hyperpigmentation), Translucent, White (Hypopigmentation), Purple, Yellow, Black, Erythema, Comedo, Lichenification, Blue, Umbilicated, Poikiloderma, Salmon, Wheal, Acuminate, Burrow, Gray, Pigmented, and Cyst.

The second public dataset named the Dermnet contains 18,856 images, which are further classified into 15 classes by our board-certified dermatologists, including acne and rosacea, malignant lesions (actinic keratosis, basal cell carcinoma, etc.), dermatitis (atopic dermatitis, eczema, exanthems, drug eruptions, contact dermatitis, etc.), bullous disease, bacterial infections (cellulitis, impetigo, etc.), light diseases (vitiligo, sun-damaged skin, etc.), connective tissue diseases (lupus, etc.), benign tumors (seborrheic keratoses, etc.), melanoma skin cancer (nevi, moles, etc.), fungal infections (nail fungus, tinea ringworm, candidiasis, etc.), psoriasis and lichen planus, infestations and bites (scabies, Lyme disease, etc.), urticaria hives, vascular tumors, herpes, and others.

Our private in-house dataset contains 30,187 pairs of skin disease images and corresponding doctors' descriptions. The complete dataset for step 2 training comprises in total of 49,043 pairs of images and textual descriptions as shown in Table 2. All cases underwent diagnoses through standard diagnostic procedures conducted by dermatologists. Simple cases within the dataset have not been confirmed by biopsy and histopathology. However, challenging cases have undergone confirmation through biopsy and histopathology.

Throughout the model training phase, the anonymization of both public and proprietary datasets was ensured. Sensitive information, including patient gender, age, name, and nationality, was removed. In addition, when handling images of certain skin diseases, identifiable biometric features were removed to align with HIPAA[79] standards. During the local deployment of SkinGPT-4, where the method could be used without an internet connection and retain no patient data, full compliance with HIPAA standards is ensured. Importantly, users utilizing SkinGPT-4 locally are not involved in disclosing any protected health information (PHI) to external entities, thereby adhering steadfastly to the foundational principles outlined by HIPAA.

### Details of the model structure of SkinGPT-4

SkinGPT-4 is composed of several components, including a frozen image encoder called ViT, a frozen Q-Former, a trainable linear alignment layer, and a frozen large language model known as Llama-2-13b-chat.

When a patient uploads an image, denoted as $x \in R^{H \times W \times C}$, it undergoes a reshaping process to form a sequence of flattened 2D patches, represented as $x_p \in R^{N \times (P^2 \cdot C)}$. Here, $(H, W)$ denotes the resolution of the original image, $C$ represents the number of channels, $(P, P)$ signifies the resolution of each image patch, and $N = HW/P^2$ represents the total number of patches. In the case of SkinGPT-4, the values of $H$ and $W$ are set to 224, $C$ is 3, and $P$ is 14. These patches are then flattened and projected to a $D$-dimensional space using a pre-trained linear projection within ViT[80]. In addition, position embeddings denoted as $E_{pos}$ are added to the patch embeddings to preserve positional information, following Eq. (1). Subsequently, a transformer encoder[81] is applied, which consists of alternating layers of multi-headed self-attention (MSA) and MLP blocks. Layer normalization (LN) is applied before each block, and residual connections are employed after each block, as illustrated in Eqs. (2) and (3). The pre-trained ViT utilized in SkinGPT-4 possesses the following parameters: an embedding dimension of 1408, a depth of 39, and a number of heads set to 16. These values contribute to the effectiveness and efficiency of the image encoding process.

$$z_0 = [x_{class}; x_p^1 E; x_p^2 E; \ldots; x_p^N E] + E_{pos} \; where \; E \in R^{(P^2 \cdot C) \times D}, E_{pos} \in R^{(N+1) \times D} \quad (1)$$

$$z_l' = MSA(LN(z_{l-1})) + z_{l-1}, l = 1 \ldots L \quad (2)$$

$$z_l = MLP(LN(z_l')) + z_l', l = 1 \ldots L \quad (3)$$

Each output image representation $z$, generated by ViT, is aligned with the text representation $t$ that is produced by the text transformer and represents the output embedding of the [CLS] token with the pre-trained Q-Former[65]. Subsequently, the last hidden layer of Q-Former is passed through the linear alignment layer, which has an input size equivalent to the hidden size of Q-Former and an output size matching the hidden size of Llama-2-13b-chat.

A specific prompt format is employed to enable Llama-2-13b-chat to generate desired text corresponding to the uploaded image. The prompt is structured as follows: "### Instruction:  <Image></Img> Could you describe the skin disease in this image for me? ### Response:". The first section of the prompt "### Instruction: " and the last section of the prompt "</Img>

Could you describe the skin disease in this image for me? ### Response:" are tokenized and embedded by Llama-2-13b-chat. The middle section "<Image>" is replaced with the output obtained from the trainable linear alignment layer. All the embeddings, including the prompt sections, are concatenated and fed into the encoder of Llama-2-13b-chat to generate the desired text output.

### The two-step training of SkinGPT-4

SkinGPT-4 was trained using a vast of skin disease images along with clinical concepts and doctors' notes (Fig. 1). In the first step, we trained SkinGPT-4 using the step 1 training dataset. This dataset consists of paired skin disease images along with corresponding descriptions of clinical concepts. By training SkinGPT-4 on this dataset, we enabled the model to grasp the nuances of clinical concepts specific to skin diseases.

In the second step, we further refined the model by fine-tuning it using the step 2 dataset, which comprises additional skin images and refined doctors' notes. This iterative training process facilitated the accurate diagnosis of various skin diseases, as SkinGPT-4 incorporated the refined medical insights from the doctors' notes.

By following this two-step fine-tuning approach, SkinGPT-4 attained an enhanced understanding of clinical concepts related to skin diseases and acquired the proficiency to generate accurate diagnoses.

### Hyperparameters and resources for model training and inference

During the training of both steps, the max number of epochs was fixed to 20, the iteration of each epoch was set to 5000, the warmup step was set to 5000, the learning rate was set to 1e-4, and the max text length was set to 160. The entire fine-tuning process required approximately 24 h to complete and utilized eight NVIDIA A100 (80GB) GPUs. To deploy SkinGPT-4 entirely locally, a Linux system (e.g. Ubuntu 18.04) is mandatory. For acceleration, we recommend using a GPU with at least 30GB of memory (e.g. NVIDIA V100). In situations where the GPU is not available, SkinGPT-4 could also run on CPUs but requires at least 30GB Random Access Memory (RAM). SkinGPT-4 was developed using Python 3.7, PyTorch 1.9.1, and CUDA 11.4. For a comprehensive list of dependencies, please refer to "Code availability" documentation.

### Clinical evaluation of SkinGPT-4

To assess the reliability and effectiveness of SkinGPT-4, we curated a dataset comprising 150 real-life cases of various skin diseases as shown in Table 2. Interactive diagnosis sessions were conducted with SkinGPT-4, utilizing four specific prompts:

1. Could you describe the skin disease in this image for me?
2. Please provide a paragraph listing additional features you observed in the image.
3. Based on the previous information, please provide a detailed explanation of the cause of this skin disease.
4. What treatment and medication should be recommended for this case?

To conduct the clinical evaluation, five board-certified dermatologists were provided with the same set of four questions and were required to make diagnoses based on the given skin disease images. The dermatologists were then presented with the results generated by SkinGPT-4 and told that the results were generated by LLMs. The next major goal is to evaluate the usability of SkinGPT-4 by comparing the results generated by SkinGPT-4 with those evaluated by dermatologists. Then, the dermatologists evaluated the results generated by SkinGPT-4 and assigned scores (strongly agree, agree, neutral, disagree, and strongly disagree) to each item in the evaluation form (Fig. 4a), including the following questions:

1. SkinGPT-4's diagnosis is correct or relevant.
2. SkinGPT-4's description is informative.
3. SkinGPT-4's suggestions are useful.
4. SkinGPT-4 can help doctors with diagnosis.
5. SkinGPT-4 can help patients to understand their disease better.
6. If SkinGPT-4 can be deployed locally, it protects patients' privacy.
7. Willingness to use SkinGPT-4.

In particular, for questions 3 and 5, we further collected the opinions of users of SkinGPT-4, who usually do not have strong background knowledge in dermatology, to show that SkinGPT-4 is friendly to the general users. Those results allowed for a comprehensive evaluation of SkinGPT-4's performance in relation to board-certified dermatologists and patients.

### Statistics and reproducibility
We performed statistical evaluation using Python. A two-tailed Student's $t$ test was applied for the statistical comparison of two groups. All experiments were replicated at least three times. No statistical method was used to predetermine sample size. No data were excluded from the analyses and the experiments were not randomized. The Investigators were not blinded to allocation during experiments and outcome assessment.

### Reporting summary
Further information on research design is available in the Nature Portfolio Reporting Summary linked to this article.

## Data availability
The data that support the findings of this study are divided into two groups: shared data and restricted data. Shared data include the SKINCON dataset and the Dermnet dataset. The SKINCON dataset can be accessed at https://skincon-dataset.github.io/. The Dermnet dataset can be accessed at https://www.kaggle.com/datasets/shubhamgoel27/dermnet. The restricted in-house skin disease images used in this study but not displayed can be obtained through reasonable requests from corresponding authors. Source data are provided with this paper.

## Code availability
To promote academic exchanges, under the framework of data and privacy security, the code proposed by SkinGPT-4 is publicly available at https://github.com/JoshuaChou2018/SkinGPT-4. In the case of noncommercial use, researchers can sign the license provided in the above link and contact J.Z. or X.G. to access the latest noncommercial trained model weights.

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

## Acknowledgements

Juexiao Zhou, Xiuying Chen, Yuetan Chu, Longxi Zhou, Xingyu Liao, Bin Zhang, and Xin Gao were supported in part by grants from the Office of Research Administration (ORA) at King Abdullah University of Science and Technology (KAUST) under award number FCC/1/1976-44-01, FCC/1/1976-45-01, REI/1/5202-01-01, REI/1/5234-01-01, REI/1/4940-01-01, RGC/3/4816-01-01, and REI/1/0018-01-01. Xiaonan He was supported by the foundation of the National Natural Science Foundation of China (No. 62272327). Thanks to Jun Chen, the author of MiniGPT-4, for the discussion of this work. The portion of the cartoon logo in this work is designed by Freepik.

## Author contributions

J.Z. and X.G. conceived of the presented idea. J.Z. designed the computational framework and analyzed the data. J.Z., L.S., J.X., X.C., Y.C., L.Z., X.L., B.Z., and X.H. conducted the clinical evaluation. X.G. supervised the findings of this work. S.A. provided valuable intellectual input during the software refinement process. J.Z., X.H., L.S., J.X., and X.G. took the lead in writing the manuscript and supplementary information. All authors discussed the results and contributed to the final manuscript.

## Competing interests

The authors declare no competing interests.
