## [Peer Review File · Nature Communications]

REVIEWER COMMENTS

Reviewer #1 (Remarks to the Author):

Pre-trained Multimodal Large Language Model Enhances Dermatological Diagnosis using SkinGPT-4

Summary

This paper introduces SkinGPT-4, the first multi-modal LLM-based interactive diagnostic system for dermatology. The authors have developed a new framework by combining a pre-trained vision transformer with an LLM called Falcon-40B-Instruct. To train SkinGPT-4, a two-step training strategy is proposed, utilizing a large collection of skin disease images.

This interactive framework enhances patient communication and education by allowing users to upload their own skin photos and receive information about their condition as well as treatment recommendations. The system underwent qualitative evaluation by certified dermatologists using 150 real-life cases. Furthermore, in order to prioritize privacy preservation, this framework allows for local deployment.

Strengths

- The use of multi-modal LLMs to enhance diagnosis in the field of dermatology indicates the novelty of this method in terms of its applications. These systems aim to address the shortage of dermatologists, improve interpretations of skin disease images, and provide patient-friendly diagnosis reports.
- In addition to the novelty of the application, the proposed two-step training strategy further enhances the method's novelty by enabling the adaptation of multimodal LLMs for specific domains.
- The paper discusses the advantages of each step in the two-step training strategy and presents supporting evidence in Figure 3 and Section 2.3.
- Previous works in related areas using LLMs have some limitations due to privacy issues. For example, ChatCAD uses the ChatGPT API, which requires users to submit sensitive text descriptions to an online API. In contrast, SkinGPT-4 overcomes this privacy concern by enabling local deployment. Based on the evaluation results provided, this feature has increased the users' willingness to use this framework.
- Compared to MiniGPT-4, SkinGPT-4 offers improved diagnosis and explanations. MiniGPT-4 was primarily pre-trained on natural images, which are irrelevant to medical images. In contrast, SkinGPT-4 performs better in this regard due to its two-step training strategy. In the initial step, SkinGPT-4 aligns visual and textual clinical concepts, enabling it to detect medical features. The second step enables SkinGPT-4 to map detected medical features to diseases. (Fig 3)

- The evaluation indicates that SkinGPT-4 performs significantly better in real-life cases of various skin diseases compared to MiniGPT-4. It was assessed on 150 real-life cases, comparing its diagnosis to that of certified dermatologists.

Weaknesses

- While SkinGPT-4's diagnosis achieves an 80% agreement with certified dermatologists, it is important to consider how the framework addresses the issues of misinformation and misdiagnosis.
- The evaluation did not provide information on the demographics of the patients involved during validation. Furthermore, the real-life cases provided in the supplementary material largely belong to a specific skin color, suggesting a bias. Consequently, this evaluation fails to capture any biases in the method, which is a weakness of this paper.
- The authors emphasize that they address privacy concerns by allowing for local deployment of the framework. However, the paper does not provide information on the feasibility of such deployments, as LLMs and vision transformers are large models that require specific computational resources for full local deployment. Additionally, the usability of the framework would be limited to patients who have personal access to sufficient computational resources.
- There are ethical concerns and contradictions between data sharing and the inclusion of images from real-life cases in the supplementary sections.

Detailed Comments

1. While the novelty in the application is commendable, I couldn't identify any architectural innovations compared to BLIP-2. Are there any specific improvements in SkinGPT-4 compared to BLIP-2?
2. As mentioned in the weakness section above, could you provide more insights into how the method works on different demographics (Performance on patients with dark skin tones)? In my opinion, it is necessary to have information on the patients' demographics, as there may be biases in the method that are not captured by the current evaluation method.
3. SkinGPT-4's ability to provide preliminary diagnoses empowers patients to make informed decisions about whether to seek in-person medical attention. This capability was highlighted as a strength of the method in the paper. However, a crucial concern arises regarding the reliability of these preliminary diagnoses. For instance, in a scenario where the framework incorrectly diagnoses a skin condition, a patient might forego a doctor's visit. As time passes, the condition could deteriorate, leading to a delayed visit when treatment becomes more challenging. How does the framework intend to address such situations, ensuring the reliability of its diagnostic recommendations?
4. In light of the described scenario, how is responsibility for the situation managed?
5. Regarding the evaluations, how many certified dermatologists were involved in the evaluation and comparison process? What type of evaluation was conducted? Were the evaluators aware that the generations are from LLMs?

6. In Fig 4a, what does the barplot indicate? Is it the percentage of skin disease cases that the dermatologists agree on, or is it the percentage of dermatologists?
7. When it comes to local deployment, how practical is it? Does it imply that the usability of the framework is restricted to patients who have access to sufficient computational resources? In my opinion, adding more information on local deployment would enhance the quality of the paper.
8. Regarding the supplementary section, is it ethical to share images of real-life cases when the identities of the patients are visible in the images? For example, cases like rosacea4, rosacea10, lupus1 etc. During a presentation, would it be better to blur certain areas in the images to preserve privacy?
9. Additionally, an ethical statement regarding the real-life cases used for evaluation is not provided. Furthermore, was the patient's consent obtained before including the images in the supplementary section?

Note on reproducibility: We attempted to run the code provided in the GitHub repository but encountered some issues during the installation and in the demo.py file.

Reviewer #2 (Remarks to the Author):

In Zhou*, He*, Sun* et al, the authors establish a generalized framework for machine learning-guided digital diagnosis of skin diseases spanning rashes as well as neoplasms. A significant innovation in this paper is the integration of deep learning models for both quantitative image analysis as well as natural language processing using LLMs to accurately diagnose clinical cases in dermatology. Another important component of the article is validation using clinical input from board certified dermatologists. The article is well-written and the figures are well made. I think this article is of significant general interest as well as of interest to the dermatology community and could be a worthwhile contribution to the literature. However, there are several major comments I would like addressed in a revised manuscript before I endorse publication.

Major constructive comments to address:

1. From Figure 4c, we can see reasonable consistency, but typically, in ML trained models, it is customary to provide other metrics such as sensitivity, specificity, positive and negative predictive value, accuracy, AUC and ROC curve etc. Can the authors comment on these metrics?
2. From Figure 4C, we see that a few cases confused the SkinGPT4 algorithm, such as eczema vs. psoriasis/LP, and connective tissue disease vs. acne and rosacea. Can the authors comment on why they think the algorithm had more difficulty with these cases?
3. Potentially related to the above question: I assume that classic cases of each diagnosis were used in the validation steps. How does the model perform with “edge” cases, such as rashes that seem indeterminate?
4. There is significant variation with patient-submitted images, especially with quality of smartphone cameras, image pre- and post-processing done on the phone software, angles, and lighting. How was the process standardized, if at all, for the submission of such photos?
5. What subset of the neoplasms and rashes included in the training dataset of this model are confirmed via biopsy and histopathology? Would be important to include this in the article.
6. To what degree does the training dataset represent adequately skin of color/darker skin (e.g. Fitzpatrick Type 4-6 skin)? Skin of color in dermatology remains poorly represented in machine learning models (and in many dermatology clinics around the world). This has been a deficiency in previous publications on ML in dermatology. Please see and cite <https://www.ncbi.nlm.nih.gov/pmc/articles/PMC9374341/>. I highly recommend that the authors add their response to this question to the discussion section of the article.
7. What measures does this pipeline contain to conform with HIPAA compliance and protect patient identity and information, given that it can be possible to trace an individual from photos of skin lesions, even if used locally?

Minor constructive comments to address:

1. Figure 3: I would suggest including just one panel (e.g. either A or B) for the figure in the main text, and put the other panel in the supplemental information. The text from the chat is too small to be easy to read on a computer or in print.
2. For several figures (e.g. Figure 1 and 3), the colors red/green or blue-green are used. I am worried about color blind readers. Would it be possible to change the color scheme away from red/green? This also makes the text in each chat bubble much harder to read as well.
3. Additional important citations from the recent literature should be included.
 - a. <https://www.nature.com/articles/s41586-023-06291-2>
 - b. <https://www.ncbi.nlm.nih.gov/pmc/articles/PMC9374341/>
 - c. <https://www.ncbi.nlm.nih.gov/pmc/articles/PMC9841357/>

Authors' Response to Reviews of NCOMMS-23-33692

Pre-trained Multimodal Large Language Model Enhances Dermatological Diagnosis using SkinGPT-4

Juexiao Zhou^{1,2,†}, Xiaonan He^{3,†,*}, Liyuan Sun^{4,†}, Jiannan Xu⁴, Xiuying Chen^{1,2}, Yuetan Chu^{1,2}, Longxi Zhou^{1,2}, Xingyu Liao^{1,2}, Bin Zhang^{1,2}, Xin Gao^{1,2,*}

¹Computer Science Program, Computer, Electrical and Mathematical Sciences and Engineering Division, King Abdullah University of Science and Technology (KAUST), Thuwal 23955-6900, Kingdom of Saudi Arabia

²Computational Bioscience Research Center, King Abdullah University of Science and Technology (KAUST), Thuwal 23955-6900, Kingdom of Saudi Arabia

³Emergency Critical Care Center, Beijing AnZhen Hospital, Affiliated to Capital Medical University, Beijing 100029, China

⁴Department of Dermatology, Beijing AnZhen Hospital, Affiliated to Capital Medical University, Beijing 100029, China

[†]These authors contributed equally.

*Corresponding author. e-mail: xin.gao@kaust.edu.sa

RC: *Reviewers' Comment*, AR: Authors' Response, Manuscript Text

1. Authors' General Responses

AR: We really appreciate Dr. Ekaterini Maria Lyras and both Reviewers' comments as well as constructive suggestions. We carefully followed all comments and made necessary changes to improve the quality and impact of our work. We believe that we have addressed the concerns raised by the reviewers to the best of our ability by either adding new results or clarifying misunderstandings, and we are confident that our research makes a valuable contribution to the field. To make the revised portions of the manuscript clear to the reviewers, we have highlighted the edited places in the manuscript with a **yellow highlight**.

Additionally, we are delighted to introduce a new feature of our method to the editor and reviewers. The initial submission of our manuscript to Nature Communications was on 27 Jul 2023. Over the past few

months, there has been rapid progress in the development of large language models (LLMs), leading to the emergence of updated and improved models within the domain of open-source LLMs. To ensure the optimal performance of our approach and minimize computational resource requirements, we have made updates by replacing the previously used LLM (Falcon-40B-Instruct) with a smaller but more easily deployable LLM (Llama-2-13b-chat). This new LLM, recognized as one of the state-of-the-art models according to several leaderboards, not only facilitates local deployment but also contributes to the overall efficiency of our method. We tested the effect of the replacement on multiple cases and are happy to conclude that it significantly reduces the required computational resource for local deployment (less GPU memory and computing time) and slightly increases performance on the diagnostic capabilities.

2. Responses to Reviewer #1

2.1. General comment

RC: *This paper introduces SkinGPT-4, the first multi-modal LLM-based interactive diagnostic system for dermatology. The authors have developed a new framework by combining a pre-trained vision transformer with an LLM called Falcon-40B-Instruct. To train SkinGPT-4, a two-step training strategy is proposed, utilizing a large collection of skin disease images.*

This interactive framework enhances patient communication and education by allowing users to upload their own skin photos and receive information about their condition as well as treatment recommendations. The system underwent qualitative evaluation by certified dermatologists using 150 real-life cases. Furthermore, in order to prioritize privacy preservation, this framework allows for local deployment.

Strengths

- *The use of multi-modal LLMs to enhance diagnosis in the field of dermatology indicates the novelty of this method in terms of its applications. These systems aim to address the shortage of dermatologists, improve interpretations of skin disease images, and provide patient-friendly diagnosis reports.*
- *In addition to the novelty of the application, the proposed two-step training strategy further enhances the method's novelty by enabling the adaptation of multimodal LLMs for specific domains.*
- *The paper discusses the advantages of each step in the two-step training strategy and presents supporting evidence in Figure 3 and Section 2.3.*
- *Previous works in related areas using LLMs have some limitations due to privacy issues. For example, ChatCAD uses the ChatGPT API, which requires users to submit sensitive text descriptions to an online API. In contrast, SkinGPT-4 overcomes this privacy concern by enabling local deployment. Based on the evaluation results provided, this feature has increased the users' willingness to use this framework.*
- *Compared to MiniGPT-4, SkinGPT-4 offers improved diagnosis and explanations. MiniGPT-4 was primarily pre-trained on natural images, which are irrelevant to medical images. In contrast, SkinGPT-4 performs better in this regard due to its two-step training strategy. In the initial step, SkinGPT-4 aligns visual and textual clinical concepts, enabling it to detect medical features. The second step enables SkinGPT-4 to map detected medical features to diseases. (Fig 3)*
- *The evaluation indicates that SkinGPT-4 performs significantly better in real-life cases of various skin diseases compared to MiniGPT-4. It was assessed on 150 real-life cases, comparing its diagnosis to that of certified dermatologists.*

Weaknesses

- *While SkinGPT-4’s diagnosis achieves an 80% agreement with certified dermatologists, it is important to consider how the framework addresses the issues of misinformation and misdiagnosis.*
- *The evaluation did not provide information on the demographics of the patients involved during validation. Furthermore, the real-life cases provided in the supplementary material largely belong to a specific skin color, suggesting a bias. Consequently, this evaluation fails to capture any biases in the method, which is a weakness of this paper.*
- *The authors emphasize that they address privacy concerns by allowing for local deployment of the framework. However, the paper does not provide information on the feasibility of such deployments, as LLMs and vision transformers are large models that require specific computational resources for full local deployment. Additionally, the usability of the framework would be limited to patients who have personal access to sufficient computational resources.*
- *There are ethical concerns and contradictions between data sharing and the inclusion of images from real-life cases in the supplementary sections.*

AR: We would like to express our heartfelt gratitude to you for dedicating valuable time to review our manuscript and for providing us with precious and constructive feedback. We recognize the importance of such feedback and sincerely appreciate your thoughtful comments and suggestions. We have revised our paper by following all of your comments. For the listed weaknesses, please refer to our detailed responses to the following major concerns.

2.2. Major concern #1

RC: *1. While the novelty in the application is commendable, I couldn’t identify any architectural innovations compared to BLIP-2. Are there any specific improvements in SkinGPT-4 compared to BLIP-2?*

AR: We appreciate your thorough review and valuable comments on our methodology. There are three major differences between SkinGPT-4 and BLIP-2:

1. **Architectural differences:** SkinGPT-4 adopts a larger state-of-the-art LLM compared to BLIP-2, which utilizes the smaller LLMs (FlanT5 and OPT). This architectural modification is aimed at improving conversational capabilities and enhancing the model’s performance in generating coherent responses to user queries.

2. **Training strategy:** The training strategy in BLIP-2 was identified by previous studies as potentially insufficient to align Vision Transformers (ViT) with LLMs, impacting text generation ability. In response

to this, SkinGPT-4 employs a two-step training strategy as detailed in the revised manuscript. Our two-step training strategy is designed to compel the model to learn medical features from skin disease images and generate coherent text responses, thereby improving overall usability.

3. **Q-Former fine-tuning:** Unlike BLIP-2, we did not fine-tune the Q-Former in SkinGPT-4. Instead, we directly used the Q-Former aligned with FlanT5 from BLIP2 and trained a single projection layer. This simpler linear layer proves sufficient for enabling the LLM to understand and generate relevant information under the context of skin disease images.

2.3. Major concern #2

RC: *2. As mentioned in the weakness section above, could you provide more insights into how the method works on different demographics (Performance on patients with dark skin tones)? In my opinion, it is necessary to have information on the patients' demographics, as there may be biases in the method that are not captured by the current evaluation method.*

AR: Thank you for the valuable comment. We appreciate the observation regarding the absence of demographic information and the potential bias in our results.

The reason for not including demographics is the lack of such information in our dataset. Both the public dataset and our proprietary data are completely anonymized, with sensitive information such as patient gender, age, name, and nationality removed to meet the requirements of privacy laws, such as Biosecurity Law in China and HIPAA in the USA. While we are unable to present statistical results for demographics, visual inspection indicates that the majority of our samples fall within Fitzpatrick I-IV skin tones. Notably, data for Fitzpatrick V-VI skin tones (dark skin tones) is underrepresented in our dataset, limiting the model's effectiveness in diagnosing skin diseases in individuals with Fitzpatrick V-VI.

Therefore, we did not focus on the diagnosis of Fitzpatrick V-VI skin tones in this study due to the following two reasons:

1. **Limited research and commonly recognized difficulties in the diagnosis of skin disease for Fitzpatrick V-VI skin tones:** Current studies on Fitzpatrick V-VI skin tones are relatively scarce, and state-of-the-art dermatological AI algorithms exhibit notably lower performance for lesions on dark skin compared to their efficacy for light-colored skin, especially in cases confirmed by biopsies. The primary challenge lies in the less conspicuous early characteristics of skin diseases for Fitzpatrick V-VI patients, making diagnosis more challenging. Consequently, individuals with darker skin tones often receive diagnoses at later stages, leading to increased morbidity, mortality, and associated costs.

2. Scarcity of Fitzpatrick V-VI data for model training: Existing datasets, such as the Diverse Dermatology Images (DDI) dataset, are insufficient for training deep learning models, particularly those based on large language models like SkinGPT-4. Currently, there is no Fitzpatrick V-VI data available in the required format to train SkinGPT-4.

We appreciate your observation regarding potential bias in our method, especially when it comes to diagnosing skin diseases for Fitzpatrick V-VI patients. Recognizing the importance of explicitly stating the limitations of SkinGPT-4, we have stated this concern as a key consideration for future research directions in our main text. We value your suggestion. Therefore, in our upcoming research, we intend to gather more data specifically representing individuals with darker skin tones and develop an independent version of SkinGPT-4 tailored to this demographic. In response to this concern, we refined our contribution claim and updated the discussion section in the main text, as outlined below:

Current research on Fitzpatrick V-VI (dark skin tones) is relatively limited, and the performance of state-of-the-art dermatological AI algorithms is notably inferior for lesions on dark skin compared to their efficacy for light-colored skin, especially in cases confirmed by biopsies. The primary challenge arises from the less conspicuous early characteristics of certain dark skin diseases, leading to a more challenging diagnosis. Consequently, individuals with darker skin tones often receive diagnoses at later stages, resulting in heightened morbidity, mortality, and associated costs. Compounding this issue is the scarcity of Fitzpatrick V-VI data, such as the Diverse Dermatology Images (DDI) dataset, which is insufficient for training deep learning models, particularly those based on LLMs such as SkinGPT-4. In this study, our dataset primarily comprises Fitzpatrick I-IV skin tones, inadvertently limiting the model's efficacy in diagnosing skin diseases in individuals with Fitzpatrick V-VI. To address this limitation, future research endeavors will involve the systematic collection of Fitzpatrick V-VI data and the targeted training of SkinGPT-4 to enhance its diagnostic capabilities for Fitzpatrick V-VI patients.

2.4. Major concern #3

RC: *3. SkinGPT-4's ability to provide preliminary diagnoses empowers patients to make informed decisions about whether to seek in-person medical attention. This capability was highlighted as a strength of the method in the paper. However, a crucial concern arises regarding the reliability of these preliminary diagnoses. For instance, in a scenario where the framework incorrectly diagnoses a skin condition, a patient might forego a doctor's visit. As time passes, the condition could deteriorate, leading to a delayed visit when treatment becomes more challenging. How does the framework intend to address such situations, ensuring the reliability of its diagnostic recommendations?*

AR: We appreciate your attention to the critical concern of addressing potential misinformation and misdiagnosis

associated with SkinGPT-4, particularly in the context of AI systems designed for medical diagnostics. Safeguarding patients is of utmost importance, and we are committed to addressing this challenge.

It is important to note that, in the real world, perfect AI systems do not and will not exist, so as perfect domain experts, errors are inevitable. Therefore, the key distinction lies in the error rate. Technically, we recognize the need for a solution that involves the implementation of a lifelong learning model in the future. This approach would allow continuous data collection and model refinement, thereby reducing the probability of errors over time, which is exactly what we are doing now.

Meanwhile, in our claim of contributions, it is crucial to note that SkinGPT-4 is not intended to replace dermatologists. Instead, it is conceived as an evolving and continuously optimized tool. SkinGPT-4 is envisioned to serve as an assistant in facilitating communication between patients and doctors. It can provide patients with additional information about skin diseases, while doctors can leverage SkinGPT-4 to aid in the diagnostic process. That is, SkinGPT-4 was meant to provide a second-opinion and second-voice to both patients and clinicians. SkinGPT-4 could enhance users' comprehension of their medical conditions, facilitate improved communication between patients and doctors, expedite the diagnostic process for dermatologists, help triage, and potentially promote human-centred care and healthcare equity in underdeveloped areas. Especially in many underdeveloped areas, misdiagnosis might not be the primary concern, because the most critical problem is the shortage of dermatologists, thus many patients find it impossible to get any medical advice. For example, a shortage of dermatologists is reported for most African countries (Namibia 0.8, Ghana 1.1, South Africa 3, Botswana 3.3 dermatologists per million population) in comparison with the rest of the world (UK 10, USA 36, Germany 65 dermatologists per million population). (Ref: <https://www.ncbi.nlm.nih.gov/pmc/articles/PMC9543359/>)

Since our work is the first study of multi-modal large language models in skin disease diagnosis, we acknowledge the importance of clear disclaimers and guidance on the software page. Specifically, we include a prominent advisory stating the necessity of adhering to medical advice, and we emphasize the recommendation for users to consult with a qualified physician for specific diagnostic results whenever possible. These precautionary measures are implemented to promote responsible use and ensure that users comprehend the potential risk of the SkinGPT-4 within a medical context.

We have updated the discussion section in the main text accordingly as below:

As LLMs-based applications like SkinGPT-4 continue to evolve and improve with the acquisition of even more reliable medical training data, the potential for significant advancements in online medical services is enormous. SkinGPT-4 could play a critical role in improving access to healthcare and

enhancing the quality of medical services for patients worldwide. It is crucial to underscore that no AI system is infallible and entirely free from misinformation and misdiagnosis. Therefore, SkinGPT-4 is not designed to replace dermatologists but rather to serve as an evolving and continuously optimized tool, functioning as an assistant in facilitating communication between patients and doctors. Our aspiration for SkinGPT-4 is to provide patients with more information about skin diseases, while also offering doctors valuable assistance in the diagnostic process. Therefore, we included clear disclaimers and guidance on the software page. This includes a prominent advisory, emphasizing the importance of adhering to medical advice, and a strong recommendation to consult with a qualified physician for specific diagnostic results. These precautionary measures are in place to encourage responsible use and ensure that users comprehend the limitations of the software in a medical context. We will continue our research in this field to further develop and refine this technology.

2.5. Major concern #4

RC: *4. In light of the described scenario, how is responsibility for the situation managed?*

AR: Thank you for this important concern. We acknowledge the importance of managing responsibility in scenarios where users may make decisions based on SkinGPT-4's recommendations.

While we strive to minimize the occurrence of misinformation by continuously improving the model's accuracy and transparency, it is also crucial for users to provide informed consent before using SkinGPT-4 and be aware of the limitations and risks associated with relying on SkinGPT-4 for medical advice. As we stated in the previous response, users will be explicitly informed about the preliminary nature of the diagnoses and are encouraged to seek professional medical guidance as early as possible to ensure responsible decision-making regarding their disease. These precautionary measures are implemented to promote responsible use and ensure that users comprehend the potential risk of the SkinGPT-4 within a medical context.

We added a disclaimer before the use of the software, users must offer consent to use SkinGPT-4 as shown below.

Demo of SkinGPT-4

This is the demo of SkinGPT-4 (version June, 2023). Upload your images and start chatting!

Disclaimer

- This SkinGPT-4 application is provided for informational purposes only and is not a substitute for professional medical advice, diagnosis, or treatment. It is not intended to replace the expertise and judgment of healthcare professionals.
- The user acknowledges and understands that the information provided by SkinGPT-4 is based on a machine learning model and may not be accurate or up-to-date. The application is not capable of providing a definitive diagnosis, and the user should consult with a qualified healthcare professional for proper medical advice.
- By using this application, the user agrees that they are solely responsible for their health and well-being. The developer and the model creators disclaim any liability for any injury, damage, or harm arising from the use of SkinGPT-4.
- If you have a medical emergency, please call your doctor or emergency services immediately.
- By clicking 'Upload' below, you acknowledge that you have read and understood this disclaimer and agree to use SkinGPT-4 at your own risk.

The interface consists of several components:

- Image Upload Area:** A dashed border box containing the text "Drop Image Here - or - Click to Upload".
- Buttons:** An orange "Upload & Start Chat" button and a grey "Restart" button.
- Settings:** Two sliders for "beam search numbers" and "Temperature", both currently set to 1.
- Chat Area:** A "SkinGPT-4" input field and a "User" input field with the placeholder text "Please upload your image first".

This site was created by King Abdullah University of Science and Technology (KAUST).
The generated results are for reference only, please follow the doctor's advice.

Users must read our disclaimer and offer consent to use SkinGPT-4.

2.6. Major concern #5

RC: *5. Regarding the evaluations, how many certified dermatologists were involved in the evaluation and comparison process? What type of evaluation was conducted? Were the evaluators aware that the generations are from LLMs?*

AR: Thank you for the valuable feedback. During the evaluation, we actively engaged five certified dermatologists. As we stated in the manuscript, the outputs generated by SkinGPT-4 are different compared to traditional classification tasks. SkinGPT-4 generates textual descriptions, encompassing medical feature details and potential judgments about skin disease types. Consequently, dermatologists were required to conduct assessments from a medical perspective. Their role involved evaluating the reliability of the answers generated by SkinGPT-4 by comparing them to the diagnoses made by dermatologists. Dermatologists knew that the results were generated by SkinGPT-4. This approach ensured a comprehensive evaluation that incorporated both medical expertise and an understanding of the capabilities and limitations of LLMs in the context of dermatological diagnoses.

We incorporated additional details in the main text as below:

To assess the reliability and effectiveness of SkinGPT-4, we **curated** a dataset comprising 150 real-life cases of various skin diseases as shown in Table 2. Interactive diagnosis sessions were conducted with SkinGPT-4, utilizing four specific prompts:

1. Could you describe the skin disease in this image for me?
2. Please provide a paragraph listing additional features you observed in the image.
3. Based on the previous information, please provide a detailed explanation of the cause of this skin disease.
4. What treatment and medication should be recommended for this case?

To conduct the clinical evaluation, **five** certified dermatologists were provided with the same set of four questions and were required to make diagnoses based on the given skin disease images. The dermatologists **were then presented with the results generated by SkinGPT-4 and told that the results were generated by LLMs. The next major goal is to evaluate the usability of SkinGPT-4 by comparing the results generated by SkinGPT-4 with those evaluated by dermatologists. Then, the dermatologists evaluated the results generated by SkinGPT-4 and assigned scores (strongly agree, agree, neutral, disagree, and strongly disagree) to each item in the evaluation form (Figure 4a), including the following questions:**

1. SkinGPT-4's diagnosis is correct or relevant.
2. SkinGPT-4's description is informative.
3. SkinGPT-4's suggestions are useful.
4. SkinGPT-4 can help doctors with diagnosis.
5. SkinGPT-4 can help patients to understand their disease better.
6. If SkinGPT-4 can be deployed locally, it protects patients' privacy.
7. Willingness to use SkinGPT-4.

2.7. Major concern #6

RC: *6. In Fig 4a, what does the barplot indicate? Is it the percentage of skin disease cases that the dermatologists agree on, or is it the percentage of dermatologists?*

AR: Thank you for the valuable comment. We apologize for any confusion resulting from our unclear writing. The barplot in Fig 4a represents the percentage of skin disease cases on which the dermatologists reached an agreement.

We updated the caption of Figure 4 in the main text as below:

Clinical evaluation of SkinGPT-4 by certified offline and online dermatologists. **a.** Questionnaire-based assessment of SkinGPT-4 by offline dermatologists. **The barplot represents the percentage of skin disease cases that the dermatologists agree on.** **b.** Response time of SkinGPT-4 compared to consulting dermatologists online. **c.** Consistency test of SkinGPT-4's responses. The x-axis indicates test samples, and the y-axis indicates the diagnostic results.

2.8. Major concern #7

RC: *7. When it comes to local deployment, how practical is it? Does it imply that the usability of the framework is restricted to patients who have access to sufficient computational resources? In my opinion, adding more information on local deployment would enhance the quality of the paper.*

AR: Thank you for the valuable comment regarding the feasibility and computational resource requirements for the local deployment of our framework. We acknowledge the need to provide more detailed information on the practical aspects of local deployment, including minimum hardware requirements and potential challenges associated with limited computational resources.

We understand that advanced GPU computing resources may not be easily accessible to general users. This is also one of the major reasons why we replaced the Falcon-40B-Instruct in the previous version with a smaller and more easily deployable LLM (Llama-2-13b-chat) in the revised version. To deploy SkinGPT-4 entirely locally, a Linux system (e.g. Ubuntu 18.04) is mandatory. For acceleration, we recommend using a GPU with at least 30GB of memory (e.g. NVIDIA V100). In situations where the GPU is not available, SkinGPT-4 could also run on CPUs but requires at least 30GB RAM, which is easier for the general user to meet. Many personal laptops could meet these requirements. This adaptability ensures broader accessibility and addresses concerns about limiting the framework’s usability to patients with sufficient computational resources.

We added more details into the section “Hyperparameters and resources for model training and inference” in the main text as below:

Hyperparameters and resources for model training and inference

During the training of both steps, the max number of epochs was fixed to 20, the iteration of each epoch was set to 5000, the warmup step was set to 5000, the learning rate was set to 1e-4, and the max text length was set to 160. The entire fine-tuning process required approximately 24 hours to complete and utilized eight NVIDIA A100 (80GB) GPUs. To deploy SkinGPT-4 entirely locally, a Linux system (e.g. Ubuntu 18.04) is mandatory. For acceleration, we recommend using a GPU with at least 30GB of memory (e.g. NVIDIA V100). In situations where the GPU is not available, SkinGPT-4 could also run on CPUs but requires at least 30GB Random Access Memory (RAM). SkinGPT-4 was developed using Python 3.7, PyTorch 1.9.1, and CUDA 11.4. For a comprehensive list of dependencies, please refer to our code availability documentation.

2.9. Major concern #8

RC: *8. Regarding the supplementary section, is it ethical to share images of real-life cases when the identities of the patients are visible in the images? For example, cases like rosacea4, rosacea10, lupus1 etc. During a presentation, would it be better to blur certain areas in the images to preserve privacy?*

AR: We acknowledge your ethical concerns regarding the inclusion of images from real-life cases in the supplementary sections. To address these concerns, we explicitly checked those images and excluded identifiable features to protect privacy. This measure aligns with ethical standards and reinforces our commitment to the responsible handling of sensitive information to meet the regulation of HIPAA.

For example, in the following images:

Response Figure 1. Figures in supplementary information before and after excluding the identifiable features.

2.10. Major concern #9

RC: *9. Additionally, an ethical statement regarding the real-life cases used for evaluation is not provided. Furthermore, was the patient's consent obtained before including the images in the supplementary section?*

AR: Thank you for the valuable comment and attention to details.

We added the "Ethics" section in the Methods section of the main text as follows:

Ethics

This study employs a deep learning methodology developed through the utilization of publicly available anonymized skin disease images, as well as anonymized in-house skin disease images sourced from hospitals. All research activities strictly adhered to established ethical regulations. The ethics vote for this study was held by the Beijing AnZhen Hospital, affiliated with Capital Medical University in Beijing, China, with ethical approval obtained under referencing ID 2024002X. The ethical approval from King Abdullah University of Science and Technology was obtained under referencing ID 23IBEC100. For publicly available anonymized skin disease images, we meticulously followed the policies and restrictions outlined by the respective datasets. Consequently, patients' informed consent was not obtained externally. For the in-house dataset, informed consent was collected from all participants whose images are featured in this study. We utilized pre-collected data from the hospital, which is a common practice in dermatological diagnosis. To uphold data anonymity, any sensitive privacy information was systematically removed from the dataset. The use of this particular subset of the data was granted a waiver for informed consent due to its anonymized nature. We did not collect any protected health information (PHI) from the patients who participated in our study, ensuring the confidentiality and privacy of their information.

2.11. Major concern #10

RC: *Note on reproducibility: We attempted to run the code provided in the GitHub repository but encountered some issues during the installation and in the demo.py file.*

AR: We appreciate your efforts in attempting to run the provided code and apologize for any issues encountered. We would like to clarify that our team is consistently updating SkinGPT-4 to adapt to the latest state-of-the-art LLMs. We apologize for any inconvenience caused to you.

The version currently available on GitHub has undergone testing on a workstation with one V100 GPU and should be functional. To use it, please follow these steps:

1. Set up the environment.

```
git clone https://github.com/JoshuaChou2018/SkinGPT-4.git

cd SkinGPT-4

conda env create -f environment.yml

conda activate skingpt4_llama2

conda install -c conda-forge mamba=1.4.7

conda install pytorch==2.0.0 torchvision==0.15.0 torchaudio==2.0.0 pytorch-cuda=11.8 -c pytorch -c nvidia
```

2. Prepare LLM weight.

```
# Make sure you have git-lfs installed (https://git-lfs.com)

git lfs install

cd SkinGPT-4-llama2/weights

git clone https://huggingface.co/meta-llama/Llama-2-13b-chat-hf
```

3. Modify line 16 at SkinGPT-4-llama2/skingpt4/configs/models/skingpt4_llama2_13bchat.yaml to be the path of Llama-2-13b-chat-hf.

4. Prepare SkinGPT-4 weight. For the moment, due to the technology transformation and privacy concerns, we are unable to publicly share the final trained model of SkinGPT-4 on both step 1 and step 2 datasets. Still, to facilitate testing, we provided the pre-trained SkinGPT-4 weights based on the base dataset only for users and reviewers to test our software. The model could be downloaded at: <https://drive.google.com/file/d/1tcwEKSBl8J7wUKBJDwptcH7AwB5Ge7iW/view?usp=sharing>

5. Modify line 10 at SkinGPT-4-llama2/eval_configs/skingpt4_eval_llama2_13bchat.yaml to be the path of SkinGPT-4 weight.

6. Execute demo.py

```
python demo.py --cfg-path eval_configs/skingpt4_eval_llama2_13bchat.yaml --gpu-id 0
```

We tested the procedure above on a new Ubuntu 18.04 system and are sure that demo.py file could be executed without error. In case the reviewer encounters any other issues, please feel free to open an issue anonymously at our code repository and we will respond immediately.

3. Responses to Reviewer #2

4. General Responses

RC: *In Zhou*, He*, Sun* et al, the authors establish a generalized framework for machine learning-guided digital diagnosis of skin diseases spanning rashes as well as neoplasms. A significant innovation in this paper is the integration of deep learning models for both quantitative image analysis as well as natural language processing using LLMs to accurately diagnose clinical cases in dermatology. Another important component of the article is validation using clinical input from board certified dermatologists. The article is well-written and the figures are well made. I think this article is of significant general interest as well as of interest to the dermatology community and could be a worthwhile contribution to the literature. However, there are several major comments I would like addressed in a revised manuscript before I endorse publication.*

AR: We would like to express our gratitude to you for dedicating valuable time to review our manuscript. We recognize the value of constructive criticism and are sincerely grateful for your insightful remarks and recommendations. We have revised our paper by following all of your comments.

4.1. Major concern #1

RC: *1. From Figure 4c, we can see reasonable consistency, but typically, in ML trained models, it is customary to provide other metrics such as sensitivity, specificity, positive and negative predictive value, accuracy, AUC and ROC curve etc. Can the authors comment on these metrics?*

AR: We appreciate your suggestion to include metrics commonly used in classical classification tasks. Figure 4a and Figure 4c in the main text could help us to conclude those metrics as shown in Response Table 1 below.

Major Classes of Skin Disease	Recall	Precision
Acne and Rosacea	0.9	1
Malignant Lesions (Actinic Keratosis, Basal Cell Carcinoma, etc.)	0.7	0.875
Dermatitis (Atopic Dermatitis, Eczema, Exanthems, Drug Eruptions, Contact Dermatitis, etc.)	1	0.769
Bullous Disease	1	1
Bacterial Infections (Cellulitis, Impetigo, etc.)	0.2	0.667
Light Diseases (vitiligo, sun damaged skin, etc.)	0.8	0.8
Connective Tissue diseases (Lupus, etc.)	0.9	0.643
Benign Tumors (Seborrheic Keratoses, etc.)	1	0.769
Melanoma Skin Cancer, Nevi, Moles	0.9	0.692
Fungal Infections (Nail Fungus, Tinea Ringworm, Candidiasis, etc.)	0.7	0.778
Psoriasis and Lichen Planus	0.7	1
Infestations and Bites (Scabies, Lyme Disease, etc.)	0.9	0.818
Urticaria Hives	0.5	0.833
Vascular Tumors	0.8	0.727
Herpes	1	0.714
Total Accuracy	0.80625	/

Response Table 1. Detailed metrics for all skin disease classes.

We did not include those metrics in the previous manuscript because our task is more like an image captioning task than a classification task. The results generated by SkinGPT-4 are textual descriptions containing medical feature descriptions and potential judgments on skin disease. Those results could also vary according to users' questions. Therefore, following the common practice in LLM evaluation, it is more meaningful to systematically evaluate the reliability of generated reports of SkinGPT-4 by engaging dermatologists in the diagnostic process, allowing them to evaluate the LLM's responses from a clinical perspective by comparing the generated results with the diagnosis made by dermatologists.

4.2. Major concern #2

RC: 2. *From Figure 4C, we see that a few cases confused the SkinGPT4 algorithm, such as eczema vs. psoriasis/LP, and connective tissue disease vs. acne and rosacea. Can the authors comment on why they think the algorithm had more difficulty with these cases?*

AR: We appreciate your insight into the challenging cases illustrated in Figure 4C.

Illustrated in the figure below is the comparison of connective tissue disease case 5 in Figure 4c, dermatomyositis, and rosacea. SkinGPT-4 exhibited uncertainty in distinguishing connective tissue diseases case 5, particularly in discerning between connective tissue disease and rosacea. In comparison, dermatomyositis is an idiopathic autoimmune connective tissue disease characterized by proximal muscle weakness and skin rashes. It is clear that the primary cause of SkinGPT-4 confusion lies in the evident overlap of certain characteristics among these diverse skin diseases.

Comparison of connective tissue diseases case 5 in Figure 4c, Dermatomyositis and Rosacea.

Meanwhile, the training of SkinGPT-4 did not explicitly account for complex skin diseases, which often involve overlapping features of multiple skin diseases, posing challenges even for dermatologists. We acknowledge this limitation and emphasize our commitment to addressing it in future work. Since real-world skin disease images may encompass various skin diseases simultaneously, forming intricate and composite cases, SkinGPT-4 could describe the medical features and potential types of skin diseases in the picture

in terms of probability. When multiple skin features are present in the same picture, SkinGPT-4 may be confused. However, it's important to note that such confusion doesn't necessarily indicate inaccuracies in SkinGPT-4 but rather highlights the presence of features potentially associated with multiple skin diseases in the same image. This further explains why we need dermatologists to evaluate the performance of SkinGPT-4 case by case. Tackling these complex scenarios constitutes a vital area for future research, and we recognize it as a key direction for further investigation.

We updated the discussion section in the main text as below:

The diagnosis of intricate skin diseases poses another challenge for SkinGPT-4. In reality, complex skin diseases frequently occur, encompassing a combination of diverse skin diseases exhibiting a multitude of characteristics. Currently, there is a lack of datasets containing multi-label skin disease images along with corresponding dermatologists' diagnoses. Addressing this gap in data constitutes a key focus for future research endeavors to apply SkinGPT-4 in the diagnosis of complex skin diseases.

4.3. Major concern #3

RC: *3. Potentially related to the above question: I assume that classic cases of each diagnosis were used in the validation steps. How does the model perform with "edge" cases, such as rashes that seem indeterminate?*

AR: Thank you for this excellent question. We recognize the importance of delving into the performance of SkinGPT-4 on "edge" cases and samples exhibiting diverse severity levels.

To address this concern, we specifically tested SkinGPT-4 on different severity levels of Acne, which were classified according to the Chinese guidelines for the treatment of acne (Revised 2019). The results of this evaluation are presented in Figure S2 in the main text (Response Figure 4). Notably, we chose to focus on Acne rather than psoriasis and atopic dermatitis. The rationale for this choice lies in the grading methodologies of the latter diseases, which typically involve assessing the percentage of skin lesions relative to the entire skin area, instead of using just one image. As demonstrated in Figure S2 in the main text (Response Figure 4), SkinGPT-4 still exhibits robust and acceptable performance when presented with skin disease images with differing severity levels of Acne.

Reference:

1. Chinese Expert Group of Acne Treatment Guidelines. Chinese guidelines for the treatment of acne (Revised 2019). *J Clin Dermatol.* 2019;48(9):583–588. doi: 10.16761/j.cnki.1000-4963.2019.09.020

We added Figure S2 to the supplementary information and updated the discussion section of the main text as below:

Deploying SkinGPT-4 in real-world scenarios may pose potential challenges, particularly due to the variability in patient-submitted images. Factors contributing to this variability include differences in smartphone camera quality, variations in image pre- and post-processing, diverse angles, and varying lighting conditions. Additionally, addressing the diverse severity levels of skin diseases presents another challenge. During the training of SkinGPT-4, we lacked the specific data required to enable the model to identify the severity of skin diseases accurately. Nevertheless, as demonstrated in Figure S2, SkinGPT-4 still exhibits robust and acceptable performance when presented with skin disease images captured under varying angles, lighting conditions, pixel densities, and resolutions with differing severity levels of Acne, which were classified according to the Chinese guidelines for the treatment of acne (Revised 2019). As shown in Figure S3, a guideline for users was also implemented, prompting them to capture images as appropriately as possible. This approach aims to standardize the format of uploaded images, facilitating SkinGPT-4's ability to identify skin disease features effectively.

Figure S2 in the supplementary information. Diagnosis generated by SkinGPT-4 on skin disease images under varying angles, lighting, pixels, and resolution conditions in Acne cases of different severity. Acne classification is proposed following the Chinese guidelines for the treatment of acne (Revised 2019).

4.4. Major concern #4

RC: *4. There is significant variation with patient-submitted images, especially with quality of smartphone cameras, image pre- and post-processing done on the phone software, angles, and lighting. How was the process standardized, if at all, for the submission of such photos?*

AR: Thank you for providing this valuable comment.

We acknowledge the potential challenges associated with variations in patient-submitted images, including differences in smartphone camera quality, image pre- and post-processing, angles, and lighting. To mitigate these challenges, SkinGPT-4 has undergone training on a diverse set of images, enhancing its robustness to various input variations. Meanwhile, the diverse set of user-uploaded images showcased in Figure S2 (Response Figure 4) reflects real-world scenarios and underscores the stability of SkinGPT-4 in handling skin disease images captured under varying angles, lighting conditions, pixel densities, and resolutions even for the same kind of skin disease.

Meanwhile, we have also incorporated user guidelines, encouraging them to capture images as appropriately as possible. This strategy is intended to standardize the format of uploaded images, thereby aiding the model in effectively identifying skin disease features, as demonstrated in Figure S3 (Response Figure 5).

Qualified:

Unqualified (Reason: lesion area is too small and not centered)

Unqualified (Reason: low light):

Unqualified (Reason: blurry picture):

Figure S3 in the supplementary information. Recommended guidelines for submitting qualified skin disease images by users when using SkinGPT-4. The image should be well-defined, allowing for clear visibility of the skin disease area, including its color, shape, and other relevant details visible to the naked eye. The skin disease area is suggested to be positioned at the center of the picture. Examples of qualified and unqualified images are provided..

We updated the discussion section in the main text as below:

Deploying SkinGPT-4 in real-world scenarios may pose potential challenges, particularly due to the variability in patient-submitted images. Factors contributing to this variability include differences in smartphone camera quality, variations in image pre- and post-processing, diverse angles, and varying lighting conditions. Additionally, addressing the diverse severity levels of skin diseases presents another challenge. During the training of SkinGPT-4, we lacked the specific data required to enable the model to identify the severity of skin diseases accurately. Nevertheless, as demonstrated in Figure S2, SkinGPT-4 still exhibits robust and acceptable performance when presented with skin disease images captured under varying angles, lighting conditions, pixel densities, and resolutions with differing severity levels of Acne, which were classified according to the Chinese guidelines for the treatment of acne (Revised 2019). As shown in Figure S3, a guideline for users was also implemented, prompting them to capture images as appropriately as possible. This approach aims to standardize the format of uploaded images, facilitating SkinGPT-4's ability to identify skin disease features effectively.

4.5. Major concern #5

RC: *5. What subset of the neoplasms and rashes included in the training dataset of this model are confirmed via biopsy and histopathology? Would be important to include this in the article.*

AR: Thank you for the valuable comment.

It is important to note that neoplasms encompass basal cell epithelioma, seborrheic keratosis, etc., while rashes include urticaria, eczema, purpura, psoriasis, etc. In clinical practice, not every diagnosis necessitates biopsy and pathology. Visible manifestations often eliminate the need for additional biopsy or histopathology.

For the public dataset we utilized, the Fitzpatrick 17k disease annotations lack verification through skin biopsy. In contrast, all DDI diseases underwent verification through skin biopsy. For Dermnet dataset, we did not find any explicit statement regarding which subset of neoplasms and rashes included in the training dataset were confirmed via biopsy and histopathology.

For our proprietary dataset, it is crucial to emphasize that our proprietary cases underwent diagnoses through standard diagnostic procedures conducted by dermatologists. Consequently, simple cases within the proprietary dataset have not been confirmed by biopsy and histopathology, as it was not deemed necessary. However, challenging cases have undergone confirmation through biopsy and histopathology.

We updated the dataset section in the main text accordingly as below:

The first public dataset named SKINCON is the first medical dataset densely annotated by domain experts to provide annotations useful across multiple disease processes. SKINCON is a skin disease dataset densely annotated by dermatologists and it includes 3230 images from the Fitzpatrick 17k skin disease dataset densely annotated with 48 clinical concepts as shown in **Table 1**, 22 of which have at least 50 images representing the concept, and 656 skin disease images from the Diverse Dermatology Images dataset. **The Fitzpatrick 17k disease annotations lack verification through skin biopsy. In contrast, all DDI diseases underwent verification through skin biopsy.** The 48 clinical concepts proposed by SKINCON include Vesicle, Papule, Macule, Plaque, Abscess, Pustule, Bulla, Patch, Nodule, Ulcer, Crust, Erosion, Excoriation, Atrophy, Exudate, Purpura/Petechiae, Fissure, Induration, Xerosis, Telangiectasia, Scale, Scar, Friable, Sclerosis, Pedunculated, Exophytic/Fungating, Warty/Papillomatous, Dome-shaped, Flat-topped, Brown (Hyperpigmentation), Translucent, White (Hypopigmentation), Purple, Yellow, Black, Erythema, Comedo, Lichenification, Blue, Umbilicated, Poikiloderma, Salmon, Wheal, Acuminate, Burrow, Gray, Pigmented, and Cyst.

Our private in-house dataset contains 30,187 pairs of skin disease images and corresponding doctors' descriptions. The complete dataset for step 2 training comprises in total of 49,043 pairs of images and textual descriptions as shown in **Table 2**. **All cases underwent diagnoses through standard diagnostic procedures conducted by dermatologists. Simple cases within the dataset have not been confirmed by biopsy and histopathology. However, challenging cases have undergone confirmation through biopsy and histopathology.**

4.6. Major concern #6

RC: *6. To what degree does the training dataset represent adequately skin of color/darker skin (e.g. Fitzpatrick Type 4-6 skin)? Skin of color in dermatology remains poorly represented in machine learning models (and in many dermatology clinics around the world). This has been a deficiency in previous publications on ML in dermatology. Please see and cite <https://www.ncbi.nlm.nih.gov/pmc/articles/PMC9374341/>. I highly recommend that the authors add their response to this question to the discussion section of the article.*

AR: We appreciate your insightful observation and we acknowledge the limitation in our training dataset. Both the public dataset and our proprietary data are completely anonymized, with sensitive information such as patient gender, age, name, and nationality removed to meet the requirements of privacy laws, such as Biosecurity Law in China and HIPAA in the USA. Therefore, we were unable to present statistical results for demographics. Visual inspection indicates that the majority of our samples fall within Fitzpatrick I-IV skin tones. Notably, data for Fitzpatrick V-VI skin tones (dark skin tones) is underrepresented in our dataset, limiting the model's

effectiveness in diagnosing skin diseases in individuals with Fitzpatrick V-VI.

Therefore, we did not focus on the diagnosis of Fitzpatrick V-VI skin tones due to the following two reasons:

- 1. Limited research and commonly recognized difficulties in the diagnosis of skin disease for Fitzpatrick V-VI skin tones:** Current studies on Fitzpatrick V-VI skin tones are relatively scarce, and state-of-the-art dermatological AI algorithms exhibit notably lower performance for lesions on dark skin compared to their efficacy for light-colored skin, especially in cases confirmed by biopsies. The primary challenge lies in the less conspicuous early characteristics of skin diseases for Fitzpatrick V-VI patients, making diagnosis more challenging. Consequently, individuals with darker skin tones often receive diagnoses at later stages, leading to increased morbidity, mortality, and associated costs.
- 2. Scarcity of Fitzpatrick V-VI data for model training:** Existing datasets, such as the Diverse Dermatology Images (DDI) dataset, are insufficient for training deep learning models, particularly those based on large language models like SkinGPT-4. Currently, there is no Fitzpatrick V-VI data available in the required format to train SkinGPT-4.

Recognizing the importance of explicitly stating the limitations of SkinGPT-4, we have stated this concern as a key consideration for future research directions in our main text. We value your suggestion. Therefore, in our upcoming research, we intend to gather more data specifically representing individuals with darker skin tones and develop an independent version of SkinGPT-4 tailored to this demographic. In response to this concern, we refined our contribution claim, cited the paper, and updated the discussion section in the main text, as outlined below:

Current research on Fitzpatrick V-VI (dark skin tones) is relatively limited, and the performance of state-of-the-art dermatological AI algorithms is notably inferior for lesions on dark skin compared to their efficacy for light-colored skin, especially in cases confirmed by biopsies. The primary challenge arises from the less conspicuous early characteristics of certain dark skin diseases, leading to a more challenging diagnosis. Consequently, individuals with darker skin tones often receive diagnoses at later stages, resulting in heightened morbidity, mortality, and associated costs. Compounding this issue is the scarcity of Fitzpatrick V-VI data, such as the Diverse Dermatology Images (DDI) dataset, which is insufficient for training deep learning models, particularly those based on LLMs such as SkinGPT-4. In this study, our dataset primarily comprises Fitzpatrick I-IV skin tones, inadvertently limiting the model's efficacy in diagnosing skin diseases in individuals with Fitzpatrick V-VI. To address this limitation, future research endeavors will involve the systematic collection of Fitzpatrick V-VI data and the targeted training of SkinGPT-4 to enhance its diagnostic capabilities for Fitzpatrick V-VI patients.

4.7. Major concern #7

RC: *7. What measures does this pipeline contain to conform with HIPAA compliance and protect patient identity and information, given that it can be possible to trace an individual from photos of skin lesions, even if used locally?*

AR: We appreciate your concern regarding patient privacy and our commitment to complying with regulations such as HIPAA.

Our institution has a strict data transfer agreement with our collaborators under the KAUST Ref. OTA1363 to ensure data safety. We also strictly adhere to the HIPAA privacy rule during both the model training and local deployment, addressing the use and disclosure of protected health information (PHI).

Throughout the model training phase, the anonymization of both public and proprietary datasets was ensured. Sensitive information, including patient gender, age, name, and nationality, was removed. Additionally, when handling images of certain skin diseases, identifiable biometric features were removed to align with HIPAA standards.

During the local deployment of SkinGPT-4, where the method could be used without an internet connection and retain no patient data, full compliance with HIPAA standards is ensured. Importantly, users utilizing SkinGPT-4 locally are not involved in disclosing any PHI to external entities, thereby adhering steadfastly to the foundational principles outlined by HIPAA.

We updated the dataset section in the main text as below:

Throughout the model training phase, the anonymization of both public and proprietary datasets was ensured. Sensitive information, including patient gender, age, name, and nationality, was removed. Additionally, when handling images of certain skin diseases, identifiable biometric features were removed to align with HIPAA standards. During the local deployment of SkinGPT-4, where the method could be used without an internet connection and retain no patient data, full compliance with HIPAA standards is ensured. Importantly, users utilizing SkinGPT-4 locally are not involved in disclosing any protected health information (PHI) to external entities, thereby adhering steadfastly to the foundational principles outlined by HIPAA.

4.8. Minor concern #1

RC: *1. Figure 3: I would suggest including just one panel (e.g. either A or B) for the figure in the main text, and put the other panel in the supplemental information. The text from the chat is too small to be easy to read on a computer or in print.*

AR: Thank you for the valuable comment and we have revised it accordingly. In the main text, we retained only panel A of Figure 3 and moved panel B to the supplementary information.

Figure 3 in main text. Diagnosis generated by SkinGPT-4, SkinGPT-4 (step 1 only), SkinGPT-4 (step 2 only), MiniGPT-4 and Dermatologists. This figure shows a case of actinic keratosis.

4.9. Minor concern #2

RC: 2. For several figures (e.g. Figure 1 and 3), the colors red/green or blue-green are used. I am worried about color-blind readers. Would it be possible to change the color scheme away from red/green? This also makes the text in each chat bubble much harder to read as well.

AR: Thank you for raising this important suggestion. We revised accordingly as below and in the previous response:

Figure 1 (in main text). Illustration of SkinGPT-4. SkinGPT-4 is the world’s first interactive dermatology diagnostic system based on multimodal large language models. To implement SkinGPT-4, we have designed a new framework that aligned a pre-trained vision transformer with a large language model named **Llama-2-13b-chat**. SkinGPT-4 was trained on a vast collection (52,929) of both public and in-house skin disease images, accompanied by clinical concepts and doctors’ notes. With SkinGPT-4, users could upload their own skin photos for diagnosis, and SkinGPT-4 could autonomously determine the characteristics and categories of skin conditions, perform analysis, provide treatment recommendations, and allow interactive diagnosis. On the right is an example of interactive diagnosis.

4.10. Minor concern #3

RC: 3. *Additional important citations from the recent literature should be included.*

a. <https://www.nature.com/articles/s41586-023-06291-2>

b. <https://www.ncbi.nlm.nih.gov/pmc/articles/PMC9374341/>

c. <https://www.ncbi.nlm.nih.gov/pmc/articles/PMC9841357/>

AR: Thank you for providing those important references. We have updated the main text below to incorporate those papers.

Nevertheless, the development of multimodal large language models for medical diagnosis is still in its early stages[51][52], particularly considering the prevalence of image-based data in the field of medical diagnosis, among which, dermatological diagnosis is a very important task but lacks relevant research on enhanced diagnosis with multimodal large language models[53][54].

References:

[52] Singhal, K., Azizi, S., Tu, T., Mahdavi, S. S., Wei, J., Chung, H. W., ... & Natarajan, V. (2023). Large language models encode clinical knowledge. *Nature*, 620(7972), 172-180.

[53] Daneshjou, R., Vodrahalli, K., Novoa, R. A., Jenkins, M., Liang, W., Rotemberg, V., ... & Chiou, A. S. (2022). Disparities in dermatology AI performance on a diverse, curated clinical image set. *Science advances*, 8(31), eabq6147.

[54] Jeong, H. K., Park, C., Henao, R., & Kheterpal, M. (2023). Deep learning in dermatology: a systematic review of current approaches, outcomes, and limitations. *JID Innovations*, 3(1), 100150.

REVIEWER COMMENTS

Reviewer #2 (Remarks to the Author):

The authors have satisfactorily addressed most of my previously mentioned major and minor concerns, as well as the concerns of the other reviewer.

There are a few other comments that I had looking at the figures and would like the authors to respond to before endorsing publication:

Figure 1 and 3 with the example of actinic keratosis: Some of the information that is generated by SkinGPT-4 as well as the text from the Dermatologist may not be factually correct or standard of care.

For example: in the last panel on Figure 3, the dermatologist says that in most cases laser treatment can treat actinic keratoses. We do not typically use lasers to treat actinic keratoses, but rather using either liquid nitrogen cryotherapy or topical chemotherapy creams such as 5-Fluorouracil. Blue light treatment (photodynamic therapy) has been used, but lasers are not the mainstay of treatment.

In the first panel on Figure 3 "SkinGPT-4" (which is similar to the panel shown in Figure 1 on the right side), it says that "actinic keratosis, also known as squamous cell carcinoma". However, actinic keratoses are precancerous lesions that have a small change of evolving into squamous cell carcinoma (or squamous cell carcinoma in situ), but by themselves are not considered cancerous and are not invasive cancers. Also, actinic keratoses do not spread to other parts of the body. Can the authors comment on this and why the program seems to generate this information?

Based on these findings, although skin-GPT seems to be accurate in terms of getting the right diagnoses of various conditions, I would like the authors to explore whether the factual information generated to the user is accurate. On my review, the factual information in the 2nd and 3rd panels of Figure 3 (although diagnoses in both are wrong), is accurate with respect to HPV infection and epidermal cysts.

Reviewer #2 (Remarks on code availability):

I have been able to download and install the software at the link provided for code availability. The website is well designed and the vignette is very helpful for local deployment.

Authors' Response to Reviews of NCOMMS-23-33692A

Pre-trained Multimodal Large Language Model Enhances Dermatological Diagnosis using SkinGPT-4

Juexiao Zhou^{1,2,3,†}, Xiaonan He^{4,†,*}, Liyuan Sun^{5,†}, Jiannan Xu⁵, Xiuying Chen^{1,2}, Yuetan Chu^{1,2}, Longxi Zhou^{1,2}, Xingyu Liao^{1,2}, Bin Zhang^{1,2}, Shawn Afvari^{3,6,7}, Xin Gao^{1,2,*}

¹Computer Science Program, Computer, Electrical and Mathematical Sciences and Engineering Division, King Abdullah University of Science and Technology (KAUST), Thuwal 23955-6900, Kingdom of Saudi Arabia.

²Computational Bioscience Research Center, King Abdullah University of Science and Technology (KAUST), Thuwal 23955-6900, Kingdom of Saudi Arabia.

³DermAssure, LLC, New York, NY, USA.

⁴Emergency Critical Care Center, Beijing AnZhen Hospital, Affiliated to Capital Medical University, Beijing 100029, China.

⁵Department of Dermatology, Beijing AnZhen Hospital, Affiliated to Capital Medical University, Beijing 100029, China.

⁶Department of Dermatology, Brigham and Women's Hospital, Harvard University, Boston, MA, USA.

⁷School of Medicine, New York Medical College, Valhalla, NY, USA.

[†]These authors contributed equally.

*Corresponding author. e-mail: xin.gao@kaust.edu.sa

RC: *Reviewers' Comment*, AR: Authors' Response, □ Manuscript Text

1. Authors' General Responses

AR: We really appreciate Dr. Ekaterini Maria Lyras and the Reviewer's comments as well as constructive suggestions. We carefully followed all comments and made necessary changes to improve the quality and impact of our work. To make the revised portions of the manuscript clear to the reviewer, we have highlighted the edited places in the manuscript with a **yellow highlight**.

2. Responses to Reviewer #2

RC: *(Remarks to the Author): The authors have satisfactorily addressed most of my previously mentioned major and minor concerns, as well as the concerns of the other reviewer.*

AR: We are grateful for your time and effort in reviewing our manuscript. Your previous comments have been instrumental in improving the quality of our work. Please find our responses to the remaining comments below.

RC: *There are a few other comments that I had looking at the figures and would like the authors to respond to before endorsing publication:*

Figure 1 and 3 with the example of actinic keratosis: Some of the information that is generated by SkinGPT-4 as well as the text from the Dermatologist may not be factually correct or standard of care.

For example: in the last panel on Figure 3, the dermatologist says that in most cases laser treatment can treat actinic keratoses. We do not typically use lasers to treat actinic keratoses, but rather using either liquid nitrogen cryotherapy or topical chemotherapy creams such as 5-Fluorouracil. Blue light treatment (photodynamic therapy) has been used, but lasers are not the mainstay of treatment.

AR: We appreciate your attention to detail and we agree with your insightful comments. Our dermatologist has also confirmed that the current recommended treatment for AK in our affiliated hospitals is the photodynamic method, though this treatment is expensive. Alternatively, cryotherapy is also a first-line treatment approach if photodynamic therapy is not viable. However, different doctors may have varying evaluations of treatment options for the same patient. Though laser treatment is not the mainstay of treatment, its immediate efficacy is notable as it allows for the removal of skin lesions on the spot. Laser therapy can be particularly advantageous for patients who find frequent clinic visits inconvenient, as it allows for longer intervals between treatments and fewer visits compared to cryotherapy.

Regarding the treatment statement for AK presented in the last panel of Figure 3, we sought advice from a board-certified dermatologist on China's largest online disease consultation platform (<https://www.alihealth.cn>). Though laser treatment may not be the primary treatment method, the assertion that "In most cases, laser treatment can effectively treat the condition." holds true. Still, we value your suggestion and to avoid presenting misleading information, we have removed the related assertion from Figure 3.

RC: *In the first panel on Figure 3 "SkinGPT-4" (which is similar to the panel shown in Figure 1 on the right side), it says that "actinic keratosis, also known as squamous cell carcinoma". However, actinic keratoses are precancerous lesions that have a small chance of evolving into squamous cell carcinoma (or squamous*

cell carcinoma in situ), but by themselves are not considered cancerous and are not invasive cancers. Also, actinic keratoses do not spread to other parts of the body. Can the authors comment on this and why the program seems to generate this information?

Based on these findings, although skin-GPT seems to be accurate in terms of getting the right diagnoses of various conditions, I would like the authors to explore whether the factual information generated to the user is accurate. On my review, the factual information in the 2nd and 3rd panels of Figure 3 (although diagnoses in both are wrong), is accurate with respect to HPV infection and epidermal cysts.

AR: Thank you for your attention to detail. We acknowledge that AK is distinct from squamous cell carcinoma, although some AK lesions have the potential to progress to squamous cell skin cancer, earning them the designation of precancers. As demonstrated in Response Figures 1-3, repeated generation by SkinGPT-4 could mitigate the likelihood of factual errors in the same case (the same AK case as shown in Figure 1 and 3 in the main text).

To answer "Can the authors comment on this and why the program seems to generate this information?", the primary goal of SkinGPT-4 is to achieve precise differential diagnosis, a task for which we personally trained the model in this study. Our results also support this goal. Additionally, we also hope SkinGPT-4 could provide accurate factual information to both patients and doctors. However, this goal is limited by the limitations inherent in LLMs, exemplified by LLaMA2 in this study. There are two potential reasons why large language models may sometimes produce factually biased descriptions. LLMs occasionally produce factually biased descriptions due to errors in the utilization of internet data during training or hallucination within the model itself. Consequently, we will develop a more specialized LLM for dermatology in future endeavors, supplanting LLaMA2 in this context. Meanwhile, with the current version of SkinGPT-4, repeated generation could mitigate the likelihood of factual errors in the same case.

Therefore, the imperative of involving dermatologists in the evaluation process is underscored. To answer "I would like the authors to explore whether the factual information generated to the user is accurate", indeed, the factual correctness of LLM-generated results has been considered in evaluating question 3, "Are SkinGPT-4's suggestions useful?" This metric primarily assesses the usefulness and factual correctness of SkinGPT-4's outputs. Previous findings indicate that 85.63% of cases are entirely factually correct.

Image

Start Chatting

Restart

beam search numbers) 5

Temperature 1

SkinGPT-4

What is actinic keratosis?

Actinic keratosis is a precancerous condition that appears as rough, scaly patches on sun-exposed areas of the skin. These patches can be tan, pink, red, or the same color as the surrounding skin. Actinic keratosis is caused by prolonged exposure to the sun or other forms of ultraviolet radiation. Treatment options include cryotherapy, topical medications, and photodynamic therapy.

Please tell us more about this skin disease, what are the causes?

Actinic keratosis is caused by prolonged exposure to the sun or other forms of ultraviolet radiation. It is a precancerous condition that can develop into squamous cell carcinoma, a type of skin cancer, if left untreated. Other risk factors for developing actinic keratosis include fair skin, light hair, and a history of sunburns.

User

Type and press Enter

This site was created by King Abdullah University of Science and Technology (KAUST).
The generated results are for reference only, please follow the doctor's advice.

Response Figure 1. The first regenerated result on the AK case.

Image

Start Chatting

Restart

beam search numbers)

Temperature

SkinGPT-4

What is actinic keratosis?

Causes and Risk Factors: Actinic keratosis is caused by prolonged exposure to the sun's ultraviolet (UV) rays, which damage the skin's DNA and cause cells to grow abnormally. People who spend a lot of time outdoors, especially during peak sun hours (10 a.m. to 4 p.m.), are more likely to develop actinic keratosis. Other risk factors include fair skin, light hair, and a history of sunburns.

Treatment and Prevention: There are several treatment options for actinic keratosis, including topical creams, cryotherapy (freezing the lesion with liquid nitrogen), and photodynamic therapy (using a light-sensitive medication and a special light source to destroy the lesion). Prevention is key, so it's essential to protect your skin from the sun by using a broad-spectrum sunscreen with an SPF of 30 or higher, wearing protective clothing, and seeking shade when the sun is strongest.

User

Type and press Enter

This site was created by King Abdullah University of Science and Technology (KAUST).
The generated results are for reference only, please follow the doctor's advice.

Response Figure 2. The second regenerated result on the AK case.

Image

Start Chatting

Restart

beam search numbers)

Temperature

SkinGPT-4

Please tell us more about this skin disease, what are the causes?

Causes and Risk Factors: As I mentioned earlier, actinic keratosis is caused by prolonged exposure to the sun's UV rays. The sun's rays can damage the skin's DNA, leading to abnormal cell growth and the development of actinic keratosis. Other risk factors include:

1. Fair skin: People with fair skin are more susceptible to actinic keratosis because their skin has less melanin, the natural pigment that protects against UV damage.
2. Light hair: People with light-colored hair have less melanin in their hair follicles, making them more prone to actinic keratosis.
3. History of sunburns: People who have had one or more sunburns are at a higher risk of developing actinic keratosis.
4. Age: Actinic keratosis is more common in older adults, as skin becomes thinner and more susceptible to UV damage with age.
5. Weakened immune system: People with weakened immune systems, such as those with autoimmune disorders or taking immunosuppressive medications, are more susceptible to actinic keratosis.

User

Type and press Enter

This site was created by King Abdullah University of Science and Technology (KAUST).
The generated results are for reference only, please follow the doctor's advice.

Response Figure 3. The third regenerated result on the AK case.

This further substantiates our assertion that SkinGPT-4 is not intended to supplant dermatologists but rather to augment their decision-making process with supplementary information for both dermatologists and patients. We also recommend users follow their doctor's advice.

The fact, "the factual information in the 2nd and 3rd panels of Figure 3 is accurate with respect to HPV infection and epidermal cysts, and the diagnoses are wrong", aligns with our expectations, because Figure 3 aims to illustrate that SkinGPT-4 trained solely on Step 1 or Step 2 data fails to accurately evaluate skin diseases. However, with our training methodology incorporating both datasets, SkinGPT-4 can achieve precise differential diagnoses.

We have revised Figures 1 and 3 in the main text with regenerated information to ensure the accuracy of the presented data. We also added a new paragraph to the discussion section as below:

The hallucination of LLMs presents another potential challenge. In the realm of medical diagnosis, the ramifications of misinformation for patients could be fatal. Given that current LLMs are trained on multiple sources, ensuring the absolute accuracy of generated medical facts is an imperative area for further investigation. Potential solutions may entail training more specialized LLMs for medical purposes and implementing iterative diagnostic generation with vote-like mechanisms. This further underscores the role of LLM-based approaches in medicine as tools meant to augment doctors' capabilities in delivering human-centered diagnoses, rather than to supplant them.

RC: *Reviewer 2 (Remarks on code availability):*

I have been able to download and install the software at the link provided for code availability. The website is well designed and the vignette is very helpful for local deployment.

AR: Thank you sincerely for dedicating your valuable time and interest to testing our software. Your feedback is greatly appreciated.

REVIEWERS' COMMENTS

Reviewer #2 (Remarks to the Author):

Thank you for addressing my comments. I endorse publication.

Authors' Response to Reviews of NCOMMS-23-33692B

Pre-trained Multimodal Large Language Model Enhances Dermatological Diagnosis using SkinGPT-4

Juexiao Zhou^{1,2,3,†}, Xiaonan He^{4,†,*}, Liyuan Sun^{5,†}, Jiannan Xu⁵, Xiuying Chen^{1,2}, Yuetan Chu^{1,2}, Longxi Zhou^{1,2}, Xingyu Liao^{1,2}, Bin Zhang^{1,2}, Shawn Afvari^{3,6,7}, Xin Gao^{1,2,*}

¹Computer Science Program, Computer, Electrical and Mathematical Sciences and Engineering Division, King Abdullah University of Science and Technology (KAUST), Thuwal 23955-6900, Kingdom of Saudi Arabia.

²Computational Bioscience Research Center, King Abdullah University of Science and Technology (KAUST), Thuwal 23955-6900, Kingdom of Saudi Arabia.

³DermAssure, LLC, New York, NY, USA.

⁴Emergency Critical Care Center, Beijing AnZhen Hospital, Affiliated to Capital Medical University, Beijing 100029, China.

⁵Department of Dermatology, Beijing AnZhen Hospital, Affiliated to Capital Medical University, Beijing 100029, China.

⁶Department of Dermatology, Brigham and Women's Hospital, Harvard University, Boston, MA, USA.

⁷School of Medicine, New York Medical College, Valhalla, NY, USA.

†These authors contributed equally.

*Corresponding author. e-mail: xin.gao@kaust.edu.sa

RC: *Reviewers' Comment*, AR: Authors' Response, Manuscript Text

1. Responses to Reviewer #2

RC: *Thank you for addressing my comments. I endorse publication.*

AR: We would like to express our heartfelt gratitude to you for dedicating valuable time to review our manuscript and for providing us with precious and constructive feedback.